# Electric polarization and magnetization in metals

Perry T. Mahon[1,2][*] and John E. Sipe[1][†]

**1** Department of Physics, University of Toronto, Toronto, Ontario M5S 1A7, Canada
**2** Department of Physics, University of Texas at Austin, Austin, TX 78712, USA

[*] perry.mahon@austin.utexas.edu , [†] sipe@physics.utoronto.ca

## Abstract

A feature of the "modern theory" is that electric polarization is not well-defined in a metallic ground state. A different approach invokes the general existence of a complete set of exponentially localized Wannier functions, with respect to which general definitions of microscopic electronic polarization and magnetization fields, and free charge and current densities are always admitted. These definitions assume no particular initial electronic state of the crystal, and the set of microscopic fields satisfy the usual relations of classical electrodynamics. Notably, when applied to a trivial insulator initially occupying its $T = 0$ ground state, the expressions for the unperturbed polarization and orbital magnetization, and for the orbital magnetoelectric polarizability tensor obtained from these different approaches can agree. However, the "modern theory of magnetization" has been extended via thermodynamic arguments to include metals and Chern insulators. We here compare with that generalization and find disagreement; the manner in which the expressions differ elucidates the distinct philosophies of these approaches. Our approach leads to the usual electrical conductivity tensor in the long-wavelength limit; in the absence of any scattering mechanisms, the dc divergence of that tensor is due to the free current density and the finite-frequency generalization of the anomalous Hall contribution arises from a combination of bound and free current densities. As well, in the limit that the electronic ground state is that of a trivial insulator, our expressions reduce to those expected for the unperturbed polarization and magnetization, and the electric susceptibility.



# 1  Introduction

In elementary classical electrodynamics, the macroscopic charge and current densities in material media are written in terms of (electric) polarization $P(x,t)$ and magnetization $M(x,t)$ fields, and "free" charge and current densities $\varrho_F(x,t)$ and $J_F(x,t)$,

$$\varrho(x,t) = -\nabla \cdot P(x,t) + \varrho_F(x,t),$$
$$J(x,t) = \frac{\partial P(x,t)}{\partial t} + c\nabla \times M(x,t) + J_F(x,t). \tag{1}$$

Going back to the time of Lorentz, $P(x,t)$ and $M(x,t)$ have typically been taken to involve those charges that remain "bound" within individual atoms and molecules, while $\varrho_F(x,t)$ and $J_F(x,t)$ are associated with other charges that are "free" to move through the medium.

In more modern treatments of metallic crystals and doped semiconductors, if the motion of the ion cores is neglected then $\varrho_F(x,t)$ and $J_F(x,t)$ are associated with intraband electronic transitions within partly occupied energy bands. In the "long-wavelength limit," where the wavelength of an applied electric field is much larger than the lattice constant, the response of those carriers is calculated as if that field were uniform. For example, in a too-simplistic model in which scattering is neglected and the relevant carriers are all assumed to have the same effective mass $m_0$ and carry the electric charge $e = -|e|$, for a uniform electric field oscillating at frequency $\omega$ with amplitude $E(\omega)$ the amplitude of the uniform current density driven in linear response is

$$J_F^{(1)}(\omega) = \frac{ie^2N}{m_0\omega}E(\omega), \tag{2}$$

where the superscript (1) indicates the linear response of the quantity, and $N$ is the density of relevant carriers.

Turning then to the other terms in the second of (1), in the long-wavelength limit the macroscopic magnetization is uniform and the "bound" current density $\partial P(t)/\partial t$ is associated with interband electronic transitions involving occupied or partly occupied energy bands. The

simplest procedure, even more elementary than a Kubo approach, is to calculate the interband absorption rate using Fermi's Golden Rule, and associate that absorption with the absorption that *would* result from a model in which the polarization responded to the electric field through a dielectric tensor $\delta^{il} + \epsilon_{\text{inter}}^{il}(\omega)$, which would give

$$P^{i(1)}(\omega) = \frac{1}{4\pi} \epsilon_{\text{inter}}^{il}(\omega) E^l(\omega), \tag{3}$$

where superscript indices indicate Cartesian components and are summed over if repeated. This association identifies the imaginary part of $\epsilon_{\text{inter}}^{il}(\omega)$, and the real part of $\epsilon_{\text{inter}}^{il}(\omega)$ can then be found using the Kramers-Kronig relation [1]. Using both (2) (or a less simplistic version) and (3), with $\epsilon_{\text{inter}}^{il}(\omega)$ so determined, a calculation of the linear response in the long-wavelength limit is complete. Sometimes one even introduces an "effective" dielectric constant $\epsilon_{\text{eff}}^{il}(\omega)$, formally writing the linear response $\boldsymbol{J}^{(1)}(t)$ of the *full* current density from the second of (1) as just $\boldsymbol{J}^{(1)}(t) = \partial \boldsymbol{P}_{\text{eff}}^{(1)}(t)/\partial t$, with

$$P_{\text{eff}}^{i(1)}(\omega) = \frac{\epsilon_{\text{eff}}^{il}(\omega) - \delta^{il}}{4\pi} E^l(\omega). \tag{4}$$

Then, in terms of the calculated $\epsilon_{\text{inter}}^{il}(\omega)$ and within the simple model (2) for the intraband response, we have

$$\epsilon_{\text{eff}}^{il}(\omega) = \delta^{il} + \epsilon_{\text{inter}}^{il}(\omega) - \frac{4\pi e^2 N}{m_0 \omega^2} \delta^{il}.$$

This strategy is somewhat indirect. One might suppose that the polarization would be defined, and then its response to the electric field calculated. But such a definition is bypassed by calculating $\epsilon_{\text{inter}}^{il}(\omega)$, that is, the contribution that a purported polarization would make to the optically induced current density. And there is no definition of either a purported polarization or magnetization that would exist before the electric field is applied.

Of course, the use of an approach that bypasses such definitions is not surprising. Any consideration of the response of "bound" charges and their currents, and the polarization and magnetization to be associated with them, is at least initially suspect from the perspective of the quantum theory of solids. In fact, problems in defining a polarization and magnetization arise even for the ground state of a crystal [2]. In recent years the "modern theories of polarization and magnetization" have been developed to clarify these concepts, primarily focused on insulators [3–5], and have provided many physical insights, including the "quantum of ambiguity" inherent to the unperturbed polarization of the $T = 0$ ground state [6], the existence of two distinct contributions to the orbital magnetization [4, 5], and that a static and uniform magnetic (electric) field can induce a polarization (magnetization) [7, 8]. However, the "modern theories" are based on static or adiabatically varying uniform fields, and are not immediately applicable to treat the optical properties of materials, especially at wavelengths so small – beyond the "long-wavelength limit" – that one has to take into account the variation of the optical fields over a unit cell.

As well, the main focus of the "modern theories" has been "topologically trivial" insulators, a class of band insulators that we define below. Indeed, among the contributions of the "modern theory of polarization" is that there may be a relationship between a certain "localization" of the electronic ground state and the polarization of an unperturbed crystal [9–11], and it has been argued that a "localized" ground state is necessary for the polarization to be well-defined; the ground state of a metallic crystal is found to violate that condition. In contrast, the "modern theory of magnetization" has been extended to include metals and Chern insulators [5, 12]. These extensions are based on thermodynamic arguments, and thus again are not

applicable to optical fields. Indeed, there seems no straightforward roadmap for extending the approach of the "modern theories" to frequency dependent polarizations, magnetizations, and free currents.

A set $\{\boldsymbol{P}(\boldsymbol{x}, t), \boldsymbol{M}(\boldsymbol{x}, t), \varrho_F(\boldsymbol{x}, t), \boldsymbol{J}_F(\boldsymbol{x}, t)\}$ that satisfies (1) is far from unique. An underlying pillar of the "modern theories" is that, in finite-sized media, $\boldsymbol{P}$ and $\boldsymbol{M}$, when taken as the usual charge and current density dipole moments, are experimentally accessible; it is implicitly assumed that the numerical value of the bulk quantities should coincide with those [4, 5, 13]. In this way, strange properties of the bulk expressions, for example, that the ground state magnetization in a Chern insulator involves a chemical potential or that polarization is not a well-defined bulk quantity in a metal while magnetization is, are justified. However, the relation between these quantities in bulk and finite-sized systems is not straightforward and considerations at the boundary are often important, even in insulators; for example, the bulk topological magnetoelectric coefficient does not generically determine that of a thin film [14, 15].

In recent work [16] we have taken a different approach, which is related more directly to the classical strategy of Lorentz, and our focus has been on bulk crystals with static ions. Here, polarization and magnetization fields, and free charge and current densities, serve as intermediary quantities that aid calculation and provide physical insight, but in general only the appropriate combinations that lead to the charge and current densities have direct physical significance.[1] To identify the electronic component of these quantities, we employ a complete set of exponentially localized Wannier functions (ELWFs)[2] (or "modified" versions thereof) with respect to which we decompose the charge and current density expectation values[3] as sums of spatially-localized contributions, one associated with each lattice site $\boldsymbol{R}$.[4] From these we define microscopic "site" polarization $\boldsymbol{p}_R^{\text{el}}(\boldsymbol{x}, t)$ and magnetization $\bar{\boldsymbol{m}}_R(\boldsymbol{x}, t)$ fields in a manner similar to that of atomic and molecular physics. Since charge continuity does not generally hold site-wise,[5] a "corrective" contribution $\tilde{\boldsymbol{m}}_R(\boldsymbol{x}, t)$ to $\bar{\boldsymbol{m}}_R(\boldsymbol{x}, t)$ arises, and in all $\boldsymbol{m}_R(\boldsymbol{x}, t) = \bar{\boldsymbol{m}}_R(\boldsymbol{x}, t) + \tilde{\boldsymbol{m}}_R(\boldsymbol{x}, t)$. Natural definitions for site charges and currents that link the lattice sites emerge as well, which are used to define microscopic free "site" charge and current densities. After identifying the ionic contribution to those site quantities, we take their lattice sums to be the microscopic polarization and magnetization fields, and free charge and current densities.

---

[1]For example, in a bulk "topologically trivial" insulator initially occupying its $T = 0$ ground state, we have previously shown [16, 17] that the electronic response to an electromagnetic field that can vary in space and time can be entirely described by the response of the site multipole moments. In contrast, in the case of a Chern insulator [39] or a $p$-doped semiconductor considered here, induced free charge and current densities are also necessary to describe the response. Although all of these quantities are gauge dependent and are therefore not experimentally accessible, that they can vanish or not nevertheless provides some insight into the physical response of the electronic degrees of freedom. In special cases the susceptibility tensors describing the response of the multipole moments can be gauge invariant and thus physically accessible; this is true of the electric susceptibility in a trivial insulator.

[2]We note that, in general, we make no assumption about the initial occupation of the electronic Bloch functions used in the construction of the ELWFs.

[3]The charge and current density operators employed within this formalism are those that arise as components of the Noether current of the Lagrangian that describes the physical system of interest, which generally involves electron field operators minimally coupled to a Maxwell electromagnetic field. Then, in general, these operators involve the electron field operators as well as the electric and magnetic Maxwell fields via vector and scalar potentials that describe them.

[4]As previously discussed [16], for the periodic systems that are the primary focus of this work, the set of "sites" – which is a non-unique collection of positions within the material medium about which localized portions of its charge and current densities might be identified – is chosen to coincide with a choice of Bravais lattice that characterizes the periodic Hamiltonian of the material medium of interest. We refer to the elements of such a set of sites as "lattice sites," and with this choice each such lattice site is itself a Bravais lattice vector.

[5]In general, for some lattice sites $\boldsymbol{R}, \boldsymbol{R}'$, the electronic site quantities $\rho_R^{\text{el}}(\boldsymbol{x}, t)$ and $\rho_{R'}^{\text{el}}(\boldsymbol{x}, t)$, and $\boldsymbol{j}_R(\boldsymbol{x}, t)$ and $\boldsymbol{j}_{R'}(\boldsymbol{x}, t)$ may have common support. Thus, in general, it may be the case that $\frac{\partial}{\partial t} \rho_R^{\text{el}}(\boldsymbol{x}, t) + \nabla \cdot \boldsymbol{j}_R(\boldsymbol{x}, t) \neq 0$, even though by construction $\frac{\partial}{\partial t} \langle \hat{\rho}(\boldsymbol{x}, t) \rangle + \nabla \cdot \langle \hat{\boldsymbol{j}}(\boldsymbol{x}, t) \rangle = 0$.

For an unperturbed "trivial" insulator occupying its $T = 0$ ground state, the spatial integral of $\bar{m}_R(x)$ ($\tilde{m}_R(x)$) coincides with that site's "atomic-like" ("itinerant") contribution to $M$ of the modern theory [5]. Here $M$ is unique (i.e. not "gauge dependent" in that it does not depend on the choice of smooth frame of the bundle of occupied electronic Hilbert spaces over the first Brillouin zone (BZ), or equivalently on how the ELWFs are chosen, assuming they are taken "occupied."); from this perspective, this is but a special case. In contrast, the spatial integral of $p_R^{el}(x)$, which coincides with that site's contribution to $P_{el}$ of the modern theory [6], is gauge dependent and only unique modulo a "quantum of ambiguity." And in the optical response of such an insulator, wherein the linearly induced charge and current densities arise entirely from induced electric and magnetic multipole moments [16], it is only the combinations of such moments corresponding to those densities that are generally gauge invariant and of direct physical significance [17]; we there find agreement with the usual approach involving a $q$-expansion of the conductivity tensor [18].

In this paper we implement this approach to treat the optical response of a metal. In this initial treatment we restrict ourselves to the long-wavelength limit and consider the independent particle approximation, in which the interaction between electrons is approximately treated through an effective potential energy characterizing the lattice and by taking the "applied" electric field in our calculations to be the macroscopic Maxwell electric field, having frequency components $E(\omega)$.

Our identification of the electronic component of the "site quantities" requires the existence of a complete set of ELWFs, which depends entirely on topological considerations [19]. In Sec. 2 we briefly discuss such issues, but the result is that if the Bloch energy eigenfunctions associated with *all* of the energy bands are employed, a complete set of ELWFs can always be constructed. Thus, the microscopic polarization and magnetization fields we introduce, and the corresponding macroscopic fields, are *always* defined, as are the macroscopic free charge and current densities. In this paper we will restrict our study to those crystalline solids for which ELWFs can be constructed from the energy eigenfunctions associated with *any* set of isolated bands – including the completely occupied and partly occupied bands in a *p*-doped semiconductor in its $T = 0$ ground state, which is the model of a metal we adopt in this first communication.

In Sec. 2 we also introduce the basic equations of our approach, relying heavily on earlier work [16]. In Sec. 3 we calculate the polarization and magnetization in the unperturbed ground state, and discuss their form; in Sec. 4 we calculate the linear response in the long-wavelength limit. If the crystal is assumed to initially possess time-reversal symmetry and its energy bands are isolated, then our results follow the pattern sketched in the first three paragraphs of this section: The induced free current can be associated with intraband transitions, and the polarization current with interband transitions. Here, of course, we have an explicit expression for the polarization and can calculate the polarization current by directly taking $\partial P(t)/\partial t$; we can thus construct an expression for $\epsilon_{eff}^{ij}(\omega)$ by direct calculation, which is gauge-invariant as expected.

The situation is more complicated in the absence of time-reversal symmetry. There we find that the first order response of each frequency component of the microscopic charge density, $\langle \hat{\rho}(x, \omega) \rangle^{(1)}$, contains a term proportional to $\omega^{-1}$ and thus diverges as $\omega \to 0$. It is then not surprising that the contribution associated with each lattice site $R$, $\rho_R^{(1)}(x, \omega)$, also diverge as $\omega \to 0$, and thus that $P^{(1)}(\omega)$ – associated with the electric dipole moments of those localized charge densities – does as well. Such a result is inevitable in the approach we adopt, where polarization and magnetization are associated with quantities localized about each lattice site. This divergent term originates from an "intraband contribution" to $P^{(1)}(\omega)$, and leads to a finite contribution to the induced macroscopic current density $-i\omega P^{(1)}(\omega)$ as $\omega \to 0$. In addition to a contribution to $J_F^{(1)}(\omega)$ that is divergent as $\omega \to 0$, which arises as an expected

generalization of (2), we also find a contribution that is finite as $\omega \to 0$. When this is combined with the contribution to $-i\omega \mathbf{P}^{(1)}(\omega)$ that is finite as $\omega \to 0$ we find a gauge-invariant contribution to $\mathbf{J}^{(1)}(\omega)$ that is finite as $\omega \to 0$, and can be identified as giving rise to the anomalous Hall current. The other contributions to the full $\mathbf{J}^{(1)}(\omega)$, which are also gauge-invariant, correspond to a generalization of (2) and to a pure interband response of $\mathbf{P}^{(1)}(\omega)$. Here we can also introduce an $\epsilon^{il}_{\mathrm{eff}}(\omega)$, but it is perhaps more natural to write

$$J^{i(1)}(\omega) = \sigma^{il}(\omega)E^l(\omega), \tag{5}$$

where $\sigma^{il}(\omega) = -i\omega(\epsilon^{il}_{\mathrm{eff}}(\omega) - \delta^{il})/(4\pi)$, and at the end of Sec. 4 we give the general expression for $\sigma^{il}(\omega)$.

A reader might ask, "why bother?" Expressions for $\sigma^{il}(\omega)$ can always be derived using Kubo's approach [20], and there is an intrinsic ambiguity in how polarization and magnetization fields are defined. And isn't the idea of a polarization – and certainly an induced polarization – in a metal suspect if it goes beyond merely the "formal" role played by, for example, the effective polarization (4)?

One reason is that usual calculations made in minimal coupling can require the identification of sum rules to show properly behaved results at low frequencies [21–24] – especially if nonlinear optical response is calculated, which is a future direction for this work – and that is not a difficulty with the calculations presented here, since the response is calculated as due to electric and magnetic fields directly. A second reason is that in an insulator there is a clear physical significance to the response of the polarization to applied fields, as has been demonstrated within the "modern theory" [6], and we feel it is interesting to see how that response can be seen to follow from the response of a $p$-doped semiconductor in the limit of vanishing doping. After all, since one can move from metal-like behavior to insulator-like behavior in this limit, it would seem physically reasonable to expect a polarization that would continuously evolve from that of a metal to that of an insulator. A third reason is that with this approach we can establish a connection to an earlier generation of calculations based on strategies introduced by Blount and co-workers.[6] A fourth reason, we feel, is the interesting way a calculation based on polarization and free currents highlights the way broken time-reversal symmetry leads to a response qualitatively different than usual. And a fifth reason is that, with its emphasis on ELWFs and the interest in those functions for electronic structure and response calculations in general, we can hope that the approach here will be useful in numerical calculations.

Our conclusions and perspectives on future work are presented in Sec. 5. Ultimately, when implemented in a metallic crystal, that our general definitions agree with past work of Blount and co-workers in a simple limit, and that the well-known $\sigma^{il}(\omega)$ results, provides positive support for our approach.

## 2 Single-particle density matrix

We consider a simple instance of a metal, a $p$-doped semiconductor, perturbed by a uniform electric field. We restrict our study to bulk crystalline solids of spatial dimension two or three ($d = 2, 3$), and implement the frozen-ion and independent-particle approximations; the spin degree of freedom is neglected. Thus, in the Heisenberg picture the only dynamical degree of freedom of the crystal is the electron field operator. In an unperturbed crystal we denote that field by $\hat{\psi}_0(\mathbf{x}, t)$ with dynamics governed by the equation of motion $i\hbar \frac{d}{dt}\hat{\psi}_0(\mathbf{x}, t) = [\hat{\psi}_0(\mathbf{x}, t), \hat{\mathsf{H}}_0]$ for one-body Hamiltonian operator $\hat{\mathsf{H}}_0 = \int \hat{\psi}_0^\dagger(\mathbf{x}, t)H_0(\mathbf{x}, \mathfrak{p}(\mathbf{x}))\hat{\psi}_0(\mathbf{x}, t)d\mathbf{x}$, where

---

[6]See Ref. [33] and references therein.

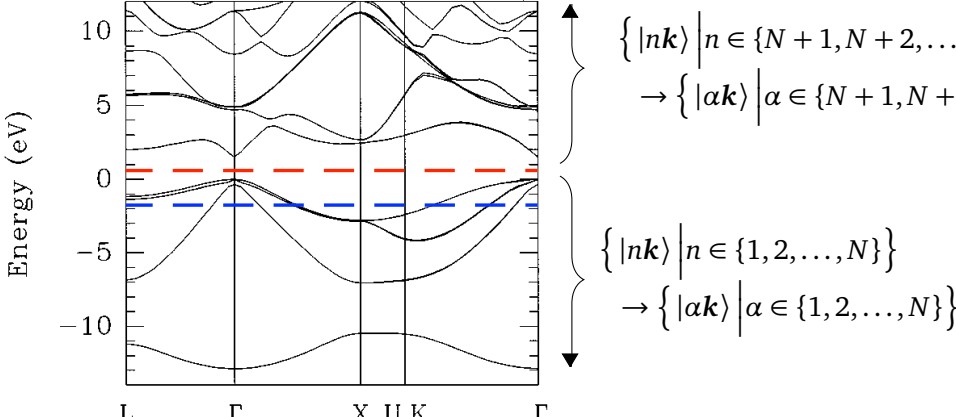

Figure 1: Schematic of the energy bands whose associated eigenvectors would be used in the construction of ELWFs in a hypothetical $d > 1$ crystalline solid (the band-structure of GaAs, which we import from a past publication [25], is used only for illustrative purposes). Upper (red) and lower (blue) horizontal dashed lines indicate possible Fermi energies for a trivial insulator and for a $p$-doped semiconductor (that we here take as a simple instance of a metal), respectively.

$$H_0\big(\boldsymbol{x}, \mathfrak{p}(\boldsymbol{x})\big) = \frac{\big(\mathfrak{p}(\boldsymbol{x})\big)^2}{2m} + V(\boldsymbol{x}), \tag{6}$$

with $V(\boldsymbol{x}) = V(\boldsymbol{x} + \boldsymbol{R})$ for any Bravais lattice vector $\boldsymbol{R}$ characterizing $H_0\big(\boldsymbol{x}, \mathfrak{p}(\boldsymbol{x})\big)$, and with

$$\mathfrak{p}(\boldsymbol{x}) \equiv \frac{\hbar}{i}\boldsymbol{\nabla} - \frac{e}{c}\boldsymbol{A}_{\text{static}}(\boldsymbol{x}), \tag{7}$$

to allow for the presence of an "internal," static, cell-periodic magnetic field described by the vector potential $\boldsymbol{A}_{\text{static}}(\boldsymbol{x})$, where $\boldsymbol{A}_{\text{static}}(\boldsymbol{x}) = \boldsymbol{A}_{\text{static}}(\boldsymbol{x} + \boldsymbol{R})$, that generally breaks time-reversal symmetry. We assume that the set of energy eigenvalues $E_{n\boldsymbol{k}}$ of the cell-periodic Hamiltonian (6) admits a band gap, below which we take the Fermi energy $E_F$ to lie, unless otherwise stated. Thus, a distinction can be made between the set of Bloch energy eigenvectors that are associated with partly occupied energy bands and those that are associated with completely unoccupied energy bands in the $T = 0$ ground state, which we take to be the initial state of the crystal.

In recent times, it has become clear that the spectral data of the relevant Hamiltonian does not entirely characterize a bulk crystal and that some topological data must also be identified. In particular, the existence of a complete set of ELWFs[7] is equivalent to the existence of a global smooth frame of the Hilbert bundle over BZ with fibres constructed point-wise for each $\boldsymbol{k} \in$ BZ as the linear span of the cell-periodic parts $u_{n\boldsymbol{k}}(\boldsymbol{x}) \equiv \langle \boldsymbol{x} | n\boldsymbol{k} \rangle$ of the Bloch energy eigenfunctions $\psi_{n\boldsymbol{k}}(\boldsymbol{x}) \equiv \langle \boldsymbol{x} | \psi_{n\boldsymbol{k}} \rangle$ associated with *all* of the energy bands, which we term the Bloch bundle.[8] In fact, such a frame always exists[9] and its components $|\alpha\boldsymbol{k}\rangle$ – which, for

---

[7]Here by "compete set" of ELWFs $W_{\alpha\boldsymbol{R}}(\boldsymbol{x}) \equiv \langle \boldsymbol{x} | \alpha\boldsymbol{R} \rangle$ we mean

$$\text{span}_{\mathbb{C}}(\{|\alpha\boldsymbol{R}\rangle \,|\, \alpha \in \{1, 2, \dots\}, \boldsymbol{R} \in \Gamma\}) \cong_{\text{Hilb}} \text{span}_{\mathbb{C}}(\{|\psi_{n\boldsymbol{k}}\rangle \,|\, \hat{\mathsf{H}}_0 \,|\psi_{n\boldsymbol{k}}\rangle = E_{n\boldsymbol{k}} \,|\psi_{n\boldsymbol{k}}\rangle\}),$$

for $\Gamma \subset \mathbb{R}^d$ a Bravais lattice of the relevant crystalline Hamiltonian.

[8]Technically, we refer to the Hilbert bundle $(\mathcal{B}, \pi, \text{BZ})$ over the first Brillouin zone $\text{BZ} \equiv \mathbb{R}^d / \Gamma^*$, for $\Gamma^*$ the dual lattice of the Hamiltonian, with fibres $\pi^{-1}(\{\boldsymbol{k}\})$ being the infinite dimensional Hilbert space spanned by $\{|n\boldsymbol{k}\rangle \,|\, n \in \mathbb{Z}\}$ as the Bloch bundle. That such a construction indeed results in a fibre bundle has been shown [40,41].

[9]See the text preceding Definition D.8 of Freed and Moore [41].

each $\boldsymbol{k} \in$ BZ, constitute an orthonormal basis of the fibre of the Bloch bundle at that $\boldsymbol{k}$ – can generally be written [19, 26, 27]

$$|\alpha\boldsymbol{k}\rangle = \sum_n U_{n\alpha}(\boldsymbol{k})|n\boldsymbol{k}\rangle \,, \tag{8}$$

where the $U_{n\alpha}(\boldsymbol{k})$ constitute a unitary matrix $U(\boldsymbol{k})$ at each $\boldsymbol{k}$; in what follows, sums are generally taken over all band indices $n$ or all "type" indices $\alpha$ unless otherwise indicated. It is then each of the $|\alpha\boldsymbol{k}\rangle$, which are smooth over the BZ, that can be mapped to an ELWF $W_{\alpha\boldsymbol{R}}(\boldsymbol{x}) \equiv \langle\boldsymbol{x}|\alpha\boldsymbol{R}\rangle$ via the (inverse) Bloch-Floquet-Zak transform [28, 29],

$$\langle\boldsymbol{x}|\alpha\boldsymbol{R}\rangle = \sqrt{\Omega_{uc}} \int_{\mathrm{BZ}} \frac{d\boldsymbol{k}}{(2\pi)^d} e^{i\boldsymbol{k}\cdot(\boldsymbol{x}-\boldsymbol{R})} \langle\boldsymbol{x}|\alpha\boldsymbol{k}\rangle \,, \tag{9}$$

where $\Omega_{uc}$ is the volume of the real space unit cell; each ELWF is identified by a type index $\alpha$ and the Bravais lattice vector $\boldsymbol{R}$ with which it is associated. Additionally, the existence of such a global smooth frame of the Bloch bundle is equivalent to the existence of a global trivialization thereof, thus any (collection of) Chern number(s) characterizing it vanish;[10] this is often understood implicitly in the physics literature.[11] The construction outlined above corresponds to using the eigenvectors associated with all of the energy bands to construct a complete set of ELWFs; in general, each $U_{n\alpha}(\boldsymbol{k})$ is nonvanishing. However, often times a number of Hilbert subbundles (of the Bloch bundle) that are associated with sets of isolated energy bands are trivial, in which case the corresponding subsets of energy eigenfunctions can be separately used to construct subsets of a complete set of ELWFs.

We here restrict our study to crystals for which the Hilbert bundle associated with *any* set of isolated energy bands is globally trivial. Taking there to be an energy gap between bands $N$ and $N+1$, $U(\boldsymbol{k})$ can always be taken of block diagonal form with the "upper left" block being $N \times N$ dimensional. If the Fermi energy lies in that gap – for example, if $E_F$ coincides with the upper (red) dashed line of Fig. 1 – then we will classify the material as a "trivial" insulator, and at each $\boldsymbol{k}$ the $U(\boldsymbol{k})$ acts on the occupied and unoccupied states $|n\boldsymbol{k}\rangle$ separately. On the other hand, if the Fermi energy lies below that gap – for example, if $E_F$ coincides with the lower (blue) dashed line of Fig. 1 – then at each $\boldsymbol{k}$ the $U(\boldsymbol{k})$ acts on the states $|n\boldsymbol{k}\rangle$ associated with the $N$ partly occupied energy bands separate from the remaining unoccupied states.

Such considerations generally apply to any crystal whose electronic spectrum has a band gap. A more general approach to generate such smooth frames in metallic crystals where it is not necessary to have isolated sets of energy bands has been formulated [30]; in future work we plan to implement this construction. However, even within the simplified scheme that we implement, an important distinction between metals and trivial insulators arises: For a trivial insulator with $N$ occupied energy bands there exists a global smooth frame of the occupied subbundle with components $|\alpha\boldsymbol{k}\rangle$ labelled by integers $\alpha \in \{1, 2, \ldots, N\}$ that satisfies $\forall \boldsymbol{k} \in \mathrm{BZ} : \mathrm{span}_{\mathbb{C}}(\{|n\boldsymbol{k}\rangle \,|E_{n\boldsymbol{k}} < E_F\}) = \mathrm{span}_{\mathbb{C}}(\{|\alpha\boldsymbol{k}\rangle \,|\alpha \in \{1, 2, \ldots, N\}\})$, while for metals with $N$ partly occupied bands there exists only such a frame that satisfies $\forall \boldsymbol{k} \in \mathrm{BZ} : \mathrm{span}_{\mathbb{C}}(\{|n\boldsymbol{k}\rangle \,|E_{n\boldsymbol{k}} < E_F\}) \subseteq \mathrm{span}_{\mathbb{C}}(\{|\alpha\boldsymbol{k}\rangle \,|\alpha \in \{1, 2, \ldots, N\}\})$ [26, 30]. This is because for metallic systems the construction of a vector bundle over BZ whose fibre at each $\boldsymbol{k} \in$ BZ is the Hilbert space spanned by the occupied $|n\boldsymbol{k}\rangle$ fails.[12] Instead, one can construct a vector bundle whose fibre at each $\boldsymbol{k}$ contains, as a subspace, the occupied Hilbert space

---

[10]This follows from the fact that there exists a global trivialization of a vector bundle if and only if the same is true of the canonical principal bundle constructed using its frames; the Chern numbers are involved in the characterization the latter. See, e.g., Proposition A.9 of [42].

[11]For case where $d = 2$, see, e.g., Eq. (1.14) of [43].

[12]Generically for metallic systems the dimensions of the occupied Hilbert subspaces associated with distinct crystal momenta $\boldsymbol{k}$ and $\boldsymbol{k}'$ differ. Thus, by definition, a vector bundle over BZ having these occupied subspaces as the fibres cannot be constructed.

at that $\boldsymbol{k}$, which yields the subset relation. Thus, while for trivial insulators the subspace $\text{span}_{\mathbb{C}}(\{|\alpha\boldsymbol{k}\rangle\,|\,\alpha\in\{1,2,\ldots,N\}\})$ contains only "ground state data," this is not so for metals. This does not pose an issue since we *do not* assert that the electronic polarization and magnetization fields involve only the initially occupied energy eigenvectors. Rather, we introduce $\boldsymbol{p}^{\text{el}}(\boldsymbol{x},t)$ and $\boldsymbol{m}(\boldsymbol{x},t)$ using any set of functions that are sufficiently localized spatially – ELWFs are the most natural and convenient choice – and, by construction, from those fields the ground state expectation values of the charge and current density operators can be found [16].

Since topological notions underlie the existence of ELWFs, the appearance of related geometric objects in many identities involving ELWFs is less opaque than it might otherwise be. One such identity that will be useful in this work is [26]

$$\int W_{\beta\boldsymbol{R}}^*(\boldsymbol{x})x^a W_{\alpha\boldsymbol{0}}(\boldsymbol{x})d\boldsymbol{x} = \frac{\Omega_{uc}}{(2\pi)^d}\int_{\text{BZ}}d\boldsymbol{k}\,e^{i\boldsymbol{k}\cdot\boldsymbol{R}}\tilde{\xi}_{\beta\alpha}^a(\boldsymbol{k}),\tag{10}$$

where, denoting the inner product on the Hilbert space spanned by the set of cell-periodic functions $u_{n\boldsymbol{k}}(\boldsymbol{x})$ (which are here taken normalized over the real-space unit cell $\Omega_{uc}\subset\mathbb{R}^d$) by $(f|g)\equiv\frac{1}{\Omega_{uc}}\int_{\Omega_{uc}}f^*(\boldsymbol{x})g(\boldsymbol{x})d\boldsymbol{x}$,

$$\tilde{\xi}_{\beta\alpha}^a(\boldsymbol{k}) = i(\beta\boldsymbol{k}|\partial_a\alpha\boldsymbol{k})\tag{11}$$

are components of the non-Abelian Berry connection that is induced by a global smooth frame with components $|\alpha\boldsymbol{k}\rangle$. Here $u_{\alpha\boldsymbol{k}}(\boldsymbol{x})\equiv\langle\boldsymbol{x}|\alpha\boldsymbol{k}\rangle$ and we adopt the shorthand $\partial_a\equiv\partial/\partial k^a$. The components (11) are related to the components of the non-Abelian Berry connection that is induced by a local smooth frame with components $|n\boldsymbol{k}\rangle$,

$$\xi_{mn}^a(\boldsymbol{k}) = i(m\boldsymbol{k}|\partial_a n\boldsymbol{k}),\tag{12}$$

via the gauge transformation

$$\sum_{\alpha\beta}U_{m\beta}(\boldsymbol{k})\tilde{\xi}_{\beta\alpha}^a(\boldsymbol{k})U_{\alpha n}^\dagger(\boldsymbol{k}) = \xi_{mn}^a(\boldsymbol{k}) + \mathcal{W}_{mn}^a(\boldsymbol{k}).\tag{13}$$

In a periodic gauge choice (8), which we always employ, $U_{m\beta}(\boldsymbol{k}) = U_{m\beta}(\boldsymbol{k}+\boldsymbol{G})$, where $\boldsymbol{G}$ is a reciprocal lattice vector, all the objects appearing here, including the Hermitian matrix $\mathcal{W}^a(\boldsymbol{k})$ [2] populated by elements

$$\mathcal{W}_{mn}^a(\boldsymbol{k}) \equiv i\sum_\alpha\big(\partial_a U_{m\alpha}(\boldsymbol{k})\big)U_{\alpha n}^\dagger(\boldsymbol{k}),\tag{14}$$

are periodic over BZ. In what follows, the $\boldsymbol{k}$-dependence of the preceding objects is usually kept implicit.

The matrix elements of $\mathcal{W}^a(\boldsymbol{k})$ that are nonvanishing depend on the structure of $U(\boldsymbol{k})$, which for the materials we consider can take the block-diagonal form discussed above. Then $\mathcal{W}_{mn}^a(\boldsymbol{k})\neq 0$ only if $m$ and $n$ lie in the same block, for if they are associated with different blocks then the values of $\alpha$ for which $U_{n\alpha}(\boldsymbol{k})\neq 0$ differ from the values of $\alpha$ for which $U_{m\alpha}(\boldsymbol{k})\neq 0$. In a trivial insulator the first, "upper left" block acts only on the occupied $|n\boldsymbol{k}\rangle$ and the second, "bottom right" block only on the unoccupied $|n\boldsymbol{k}\rangle$. If we introduce Fermi filling factors $f_{n\boldsymbol{k}} = 1$ (0) if the state $|n\boldsymbol{k}\rangle$ is initially occupied (unoccupied), then for the trivial insulator $f_{n\boldsymbol{k}} = f_n$, depending only on the band, and $\mathcal{W}_{mn}^a(\boldsymbol{k})\neq 0$ only if $f_m = f_n$. But in a $p$-doped semiconductor, which is our simple instantiation of a metal in this paper, the first block also acts on some unoccupied $|n\boldsymbol{k}\rangle$, and so in general we can have $\mathcal{W}_{mn}^a(\boldsymbol{k})\neq 0$ even if $f_{m\boldsymbol{k}}\neq f_{n\boldsymbol{k}}$.

We account for the interaction between the electron field and the "applied" electromagnetic field via the usual minimal coupling prescription. From the resulting minimal coupling

Hamiltonian, the field-theoretic charge and current density operators constituting the Noether current can be found in the usual way.[13] Associated with each spatial component of this current density operator is the differential operator

$$J_{\mathrm{mc}}^a\big(\mathbf{x}, \mathfrak{p}_{\mathrm{mc}}(\mathbf{x}, t)\big) = \frac{e}{m}\mathfrak{p}_{\mathrm{mc}}^a(\mathbf{x}, t), \tag{15}$$

where $m$ is the electron mass,

$$\mathfrak{p}_{\mathrm{mc}}(\mathbf{x}, t) \equiv \mathfrak{p}(\mathbf{x}) - \frac{e}{c}\mathbf{A}(\mathbf{x}, t), \tag{16}$$

and where the vector and scalar potentials $\mathbf{A}(\mathbf{x}, t)$ and $\phi(\mathbf{x}, t)$ describe the classical applied electromagnetic field. As a consequence, another useful identity will be

$$\int \psi_{n'\mathbf{k}'}^*(\mathbf{x})\mathfrak{p}^a(\mathbf{x})\psi_{n\mathbf{k}}(\mathbf{x})d\mathbf{x} = \mathfrak{p}_{n'n}^a(\mathbf{k})\delta(\mathbf{k} - \mathbf{k}'),$$

where the matrix elements are found to be [31]

$$\mathfrak{p}_{n'n}^a(\mathbf{k}) = \delta_{n'n}\frac{m}{\hbar}\partial_a E_{n\mathbf{k}} + \frac{im}{\hbar}\big(E_{n'\mathbf{k}} - E_{n\mathbf{k}}\big)\xi_{n'n}^a(\mathbf{k}). \tag{17}$$

Under the frozen-ion approximation implemented here, we take the positively charged ion cores that compose the underlying crystal structure of the material to be fixed, even in the presence of an applied electromagnetic field, and introduce the charge density $\rho^{\mathrm{ion}}(\mathbf{x})$ to describe the periodic distribution of these static charges. We take $\rho^{\mathrm{ion}}(\mathbf{x})$ such that the crystal as a whole is electrically neutral.

In this work, we implement a previously developed formalism [16] restricted to the "long-wavelength limit," wherein we take the applied electric field to be uniform and the magnetic field to vanish. To simplify these initial considerations we here neglect local field corrections, which can be important [32], and take the applied electric field be the macroscopic Maxwell field denoted $\mathbf{E}(t)$ [17]. Consideration of phenomena related to spatially-varying electromagnetic fields is left for future work. A quantity central to this formalism is the so-called (electronic) single-particle density matrix $\eta_{\alpha\mathbf{R}'';\beta\mathbf{R}'}(t)$, the definition of which (see Eq. (15, 27, 30, 33, 36) of Mahon *et al.* [16]) involves a generalized Peierls phase, the fermionic operators that generate ELWFs $W_{\alpha\mathbf{R}}(\mathbf{x})$ and "modified" versions $\bar{W}_{\alpha\mathbf{R}}(\mathbf{x}, t)$ thereof, and the initial electronic state of the unperturbed crystal, here taken to be the $T = 0$ ground state. The operators $\hat{a}_{n\mathbf{k}}$ and $\hat{a}_{n\mathbf{k}}^\dagger$ generating the eigenvectors $|\psi_{n\mathbf{k}}\rangle$ of the unperturbed Hamiltonian $\hat{\mathsf{H}}_0$ are also relevant in perturbative calculations and we take $|\psi_{n\mathbf{k}}\rangle \equiv \hat{a}_{n\mathbf{k}}^\dagger |\mathrm{vac}\rangle$, where $\hat{\mathsf{H}}_0 |\psi_{n\mathbf{k}}\rangle = E_{n\mathbf{k}}|\psi_{n\mathbf{k}}\rangle$. Still in the Heisenberg picture, electronic field operators $\hat{\psi}(\mathbf{x}, t)$ here evolve as $i\hbar\frac{d}{dt}\hat{\psi}(\mathbf{x}, t) = [\hat{\psi}(\mathbf{x}, t), \hat{\mathsf{H}}(t)]$, where $\hat{\mathsf{H}}(t)$ involves $\mathbf{E}(t)$.

In what follows, we account for the effect of the applied electric field perturbatively. Thus we assume the existence of a valid expansion of all electronic quantities in powers of $\mathbf{E}(t)$. In particular, we take $\eta_{\alpha\mathbf{R}'';\beta\mathbf{R}'}(t)$ of the form

$$\eta_{\alpha\mathbf{R}'';\beta\mathbf{R}'}(t) = \eta_{\alpha\mathbf{R}'';\beta\mathbf{R}'}^{(0)} + \eta_{\alpha\mathbf{R}'';\beta\mathbf{R}'}^{(1)}(t) + \dots,$$

where the superscript (0) denotes the contribution to a quantity that is independent of $\mathbf{E}(t)$, the superscript (1) denotes the contribution that is linear in $\mathbf{E}(t)$, and "..." denotes non-linear contributions, which we here neglect. In Appendices A and B we find

$$\eta_{\alpha\mathbf{R}'';\beta\mathbf{R}'}^{(0)} = \Omega_{uc}\int_{\mathrm{BZ}}\frac{d\mathbf{k}}{(2\pi)^d}e^{i\mathbf{k}\cdot(\mathbf{R}''-\mathbf{R}')}\sum_n f_{n\mathbf{k}}U_{\alpha n}^\dagger U_{n\beta}, \tag{18}$$

---

[13]See, e.g., [44].

and implementing the usual Fourier series analysis,

$$g(t) \equiv \sum_\omega e^{-i\omega t} g(\omega), \tag{19}$$

we also find that the first-order perturbative modification to $\eta_{\alpha R'';\beta R'}(t)$ due to $E(\omega)$ is

$$\eta^{(1)}_{\alpha R'';\beta R'}(\omega) = eE^l(\omega)\Omega_{uc} \int_{\mathrm{BZ}} \frac{d\boldsymbol{k}}{(2\pi)^d} e^{i\boldsymbol{k}\cdot(R''-R')} \sum_{mn} \frac{f_{nm,\boldsymbol{k}} U^\dagger_{\alpha m} \xi^l_{mn} U_{n\beta}}{E_{m\boldsymbol{k}} - E_{n\boldsymbol{k}} - \hbar(\omega + i0^+)}$$

$$- ie\frac{E^l(\omega)}{\hbar(\omega + i0^+)}\Omega_{uc} \int_{\mathrm{BZ}} \frac{d\boldsymbol{k}}{(2\pi)^d} e^{i\boldsymbol{k}\cdot(R''-R')} \sum_n (\partial_l f_{n\boldsymbol{k}}) U^\dagger_{\alpha n} U_{n\beta}, \tag{20}$$

where $f_{nm,\boldsymbol{k}} \equiv f_{n\boldsymbol{k}} - f_{m\boldsymbol{k}}$. The first term of (20) is the straight-forward generalization of the previously found perturbative modification for trivial insulators, and can be understood in the context of time-dependent perturbation theory as arising from the interaction term[14]

$$-eE^a(t) \int_{\mathrm{BZ}} d\boldsymbol{k} \sum_{n\neq m} \hat{a}^\dagger_{n\boldsymbol{k}}(t) \xi^a_{nm}(\boldsymbol{k}) \hat{a}_{m\boldsymbol{k}}(t).$$

Due to the form of this interaction term we will later describe any first-order modifications that involve the first term of (20) as being "interband." The second term of (20) is a new contribution, here related to the presence of a Fermi surface. Notably this term diverges in the dc limit, and indeed it is this term that will lead to the expected dc divergence of the induced free current density, as we later show. This term can here be understood as arising from the interaction term

$$-eE^a(t) \int_{\mathrm{BZ}} d\boldsymbol{k} \sum_n \left( \hat{a}^\dagger_{n\boldsymbol{k}}(t) \xi^a_{nn}(\boldsymbol{k}) \hat{a}_{n\boldsymbol{k}}(t) + \frac{i}{2}\hat{a}^\dagger_{n\boldsymbol{k}}(t)(\partial_a \hat{a}_{n\boldsymbol{k}}(t)) - \frac{i}{2}(\partial_a \hat{a}^\dagger_{n\boldsymbol{k}}(t))\hat{a}_{n\boldsymbol{k}}(t) \right).$$

The first contribution (that involving $\xi^a_{nn}$) to this interaction term gives a vanishing contribution to $\eta^{(1)}_{\alpha R'';\beta R'}(\omega)$ for both metals and trivial insulators initially occupying their $T = 0$ electronic ground state (see Appendix B). In contrast, although the second and third contributions as well give vanishing contributions to $\eta^{(1)}_{\alpha R'';\beta R'}(\omega)$ for trivial insulators, they give rise to finite contributions if the crystal is metallic; these finite contributions involve only those occupied $|n\boldsymbol{k}\rangle$ with energies "near" the Fermi energy. Due to the form of this interaction term we will later describe the first-order modifications that involve the second term of (20) as being "intraband." Such an identification of interaction terms that give rise to inter- and intraband contributions at linear response is implicit in the earlier works of Blount [33] and others.[15] The primary difference between our approach and those is that this investigation is a limiting case of a more general framework within which spatial and temporal variation of electric and magnetic fields can be taken into account; that is not the case in earlier works.

The limit of a trivial insulator can be reached from (18) by taking $f_{n\boldsymbol{k}} \to f_n$ and requiring each $|\alpha\boldsymbol{k}\rangle$ to be an element of either the initially occupied or unoccupied Hilbert subspace; the second condition implies that, in general, $\mathcal{W}^i_{nm}(\boldsymbol{k}) \neq 0$ only if $f_n = f_m$ (see discussion below (14)). In this limit, one can define an analogous filling factor $f_\alpha$ associated with $|\alpha\boldsymbol{k}\rangle$; we set the $f_\alpha$ associated with $|\alpha\boldsymbol{k}\rangle$ to equal the $f_n$ associated with the set of $|n\boldsymbol{k}\rangle$ used in its construction. The sum over $n$ in (18) then corresponds to the matrix multiplication of $U(\boldsymbol{k})$

---

[14]The operators $\hat{a}_{n\boldsymbol{k}}(t)$ and $\hat{a}^\dagger_{n\boldsymbol{k}}(t)$ appearing in interaction terms evolve in the interaction picture. For details, see Appendix B.

[15]See, e.g., [34], [22], and references therein.

and its inverse giving the unit matrix at each $\boldsymbol{k}$, which in components is $\delta_{\alpha\beta}$. It then follows that, in this limit,

$$\eta^{(0;\text{insulator})}_{\alpha\boldsymbol{R}'';\beta\boldsymbol{R}'} = f_\alpha \delta_{\alpha\beta}\delta_{\boldsymbol{R}''\boldsymbol{R}'},$$

as expected [16]. Implementing this limit in (20) we find

$$\eta^{(1;\text{insulator})}_{\alpha\boldsymbol{R}'';\beta\boldsymbol{R}'}(\omega) = eE^l(\omega)\Omega_{uc}\int_{\text{BZ}}\frac{d\boldsymbol{k}}{(2\pi)^d}e^{i\boldsymbol{k}\cdot(\boldsymbol{R}''-\boldsymbol{R}')}\sum_{mn}\frac{f_{nm}U^\dagger_{\alpha m}\xi^l_{mn}U_{n\beta}}{E_{m\boldsymbol{k}}-E_{n\boldsymbol{k}}-\hbar(\omega+i0^+)},$$

again, as expected [16].

## 3 Dipole moments

In general, the presence of an applied electromagnetic field will break the discrete translational symmetry of the unperturbed crystal. Consequently, in the minimally-coupled system, contributions to a given electric or magnetic multipole moment that are associated with distinct lattice sites of the crystal will generally differ. However, in the long-wavelength limit, all of the site contributions to a given multipole moment are equivalent. In particular, in this limit the site electric and magnetic dipole moments satisfy

$$\boldsymbol{\mu}_{\boldsymbol{R}}(t) = \boldsymbol{\mu}_{\boldsymbol{R}'}(t),\qquad \boldsymbol{\nu}_{\boldsymbol{R}}(t) = \boldsymbol{\nu}_{\boldsymbol{R}'}(t),$$

for any Bravais lattice vectors $\boldsymbol{R}$ and $\boldsymbol{R}'$ of $H_0\big(\boldsymbol{x},\mathfrak{p}(\boldsymbol{x})\big)$. It follows that the macroscopic polarization and magnetization fields are uniform [17] and can be written as

$$P(t) = \frac{\boldsymbol{\mu}_{\boldsymbol{R}}(t)}{\Omega_{uc}},\qquad M(t) = \frac{\boldsymbol{\nu}_{\boldsymbol{R}}(t)}{\Omega_{uc}},\tag{21}$$

for any such $\boldsymbol{R}$. Because we consider the ionic cores within the crystal to be fixed, these charges do not contribute to the magnetization; there will however be a static contribution to the polarization that is found from the "site" polarization fields that are defined from the constituents of a decomposition of $\rho^{\text{ion}}(\boldsymbol{x})$ into "site" contributions [16, 17]. Irrespective of the initial electronic state of the crystal, the electric dipole moment associated with lattice site $\boldsymbol{R}$ is defined to be

$$\mu^i_{\boldsymbol{R}}(t) \equiv \sum_{\alpha\beta\boldsymbol{R}'\boldsymbol{R}''}\left(\int(x^i-R^i)\rho_{\beta\boldsymbol{R}';\alpha\boldsymbol{R}''}(\boldsymbol{x},\boldsymbol{R};t)d\boldsymbol{x}\right)\eta_{\alpha\boldsymbol{R}'';\beta\boldsymbol{R}'}(t)+\big(\mu^{\text{ion}}_{\boldsymbol{R}}\big)^i,\tag{22}$$

(see Eq. (42, 44, 55, 57) of Mahon *et al.* [16]) and the magnetic dipole moment associated with $\boldsymbol{R}$ to be

$$\nu^i_{\boldsymbol{R}}(t) \equiv \frac{1}{2c}\sum_{\alpha\beta\boldsymbol{R}'\boldsymbol{R}''}\left(\epsilon^{iab}\int(x^a-R^a)\big(j^b_{\beta\boldsymbol{R}';\alpha\boldsymbol{R}''}(\boldsymbol{x},\boldsymbol{R};t)+\tilde{j}^b_{\beta\boldsymbol{R}';\alpha\boldsymbol{R}''}(\boldsymbol{x},\boldsymbol{R};t)\big)d\boldsymbol{x}\right)\eta_{\alpha\boldsymbol{R}'';\beta\boldsymbol{R}'}(t),\tag{23}$$

(see also Eq. (64, 66, 67) of Mahon *et al.* [16]), where $\rho_{\beta\boldsymbol{R}';\alpha\boldsymbol{R}''}(\boldsymbol{x},\boldsymbol{R};t)$, $\boldsymbol{j}_{\beta\boldsymbol{R}';\alpha\boldsymbol{R}''}(\boldsymbol{x},\boldsymbol{R};t)$, and $\tilde{\boldsymbol{j}}_{\beta\boldsymbol{R}';\alpha\boldsymbol{R}''}(\boldsymbol{x},\boldsymbol{R};t)$ are termed generalized (electronic) "site-quantity matrix elements" and were introduced previously [16]. Again, with the assumption of a valid perturbative expansion (22,23) can be written

$$\boldsymbol{\mu}_{\boldsymbol{R}}(t) = \boldsymbol{\mu}^{(0)}_{\boldsymbol{R}} + \boldsymbol{\mu}^{(1)}_{\boldsymbol{R}}(t) + \dots,$$
$$\boldsymbol{\nu}_{\boldsymbol{R}}(t) = \boldsymbol{\nu}^{(0)}_{\boldsymbol{R}} + \dots,$$

and the same can be done for (21). The first term in each such expansion is identified as the "unperturbed contribution," and these are the focus of the rest of this section.

Although the matrix elements appearing in (22,23) are generally of quantities arising from a minimally coupled Hamiltonian and written in a basis of "modified" Wannier functions [16], we explicitly show in the following subsections (and in Sec. 4) that, as usual, terms appearing at each order in a perturbative expansion can be written in terms of energy eigenvectors (or equivalently in terms of ELWFs in a crystalline solid) of the unperturbed system. When practical, we include the expression for a quantity written as a product of a BZ integral and an integral involving ELWFs, and as a single BZ integral. Although both forms are equivalent, the numerical implementation of the expressions might favor a particular form. For example, when written as a single BZ integral, some quantities involve diagonal matrix elements of the Berry connection $\xi_{nn}^a(\boldsymbol{k})$, which are typically not easy to evaluate numerically. For such quantities, evaluating integrals involving ELWFs may be more tractable.

## 3.1 Electric polarization

From (22), the electronic contribution $\mu_{\boldsymbol{R}}^{\mathrm{el}(0)}$ to $\mu_{\boldsymbol{R}}^{(0)}$ is

$$\mu_{\boldsymbol{R}}^{\mathrm{el}(0)} = e\mathrm{Re}\sum_{\alpha\beta\boldsymbol{R}'}\left(\int W_{\beta\boldsymbol{0}}^*(\boldsymbol{y})\boldsymbol{y}W_{\alpha\boldsymbol{R}'-\boldsymbol{R}}(\boldsymbol{y})d\boldsymbol{y}\right)\eta_{\alpha\boldsymbol{R}';\beta\boldsymbol{R}}^{(0)},$$

and implementing (10,13,18) we find $\mu_{\boldsymbol{R}}^{\mathrm{el}(0)}$ to be independent of $\boldsymbol{R}$, as expected. The resulting unperturbed polarization is

$$P^{i(0)} = e\int_{\mathrm{BZ}}\frac{d\boldsymbol{k}}{(2\pi)^d}\sum_n f_{n\boldsymbol{k}}\left(\xi_{nn}^i + \mathcal{W}_{nn}^i\right) + \frac{\left(\mu_{\boldsymbol{R}}^{\mathrm{ion}}\right)^i}{\Omega_{uc}}, \tag{24}$$

which is formally similar to that of a trivial insulator [6,31], for which $f_{n\boldsymbol{k}} \to f_n$. As is the case there, apart from a gauge dependent contribution, $\boldsymbol{P}^{(0)}$ vanishes if the unperturbed Hamiltonian is inversion symmetric; as discussed previously [31], we take the gauge dependence of the electronic quantities to be contained entirely within the $U(\boldsymbol{k})$ and consider any terms that involve this object, including the $\mathcal{W}^i(\boldsymbol{k})$, to be "gauge dependent." While it appears that the gauge dependent term appearing in (24) no longer generally evaluates to an element of a set of discrete values, at least not following from the same argument that is presented for trivial insulators [6], $\boldsymbol{P}^{(0)}$ maintains the physically sensible characteristic that upon shifting the origin of all ELWFs by a constant Bravais lattice vector $\boldsymbol{R}_s$, the polarization is altered by an additive constant that is proportional to $\boldsymbol{R}_s$. That is, although there is no longer a "quantum of ambiguity" associated with $\boldsymbol{P}^{(0)}$ for a general change in $U_{n\alpha}(\boldsymbol{k})$, as occurs for a trivial insulator [6], taking $|\alpha\boldsymbol{R}\rangle \to |\alpha\boldsymbol{R}+\boldsymbol{R}_s\rangle$, or equivalently $U_{n\alpha}(\boldsymbol{k}) \to e^{-i\boldsymbol{k}\cdot\boldsymbol{R}_s}U_{n\alpha}(\boldsymbol{k})$ and thus $\mathcal{W}_{nm}^a(\boldsymbol{k}) \to \mathcal{W}_{nm}^a(\boldsymbol{k}) + \delta_{nm}R_s^a$, yields

$$\boldsymbol{P}^{(0)} \to \boldsymbol{P}^{(0)} + eN_{\mathrm{el}}\boldsymbol{R}_s,$$

where $N_{\mathrm{el}} = \int_{\mathrm{BZ}}\frac{d\boldsymbol{k}}{(2\pi)^d}\sum_n f_{n\boldsymbol{k}}$ is the number of electrons per unit volume. Thus, with respect to simple shifts in the positions of the Wannier function a discrete ambiguty does arise. We do note however that an expression formally similar to (24) arises in the case of a Chern insulator [13], and while in that case the gauge dependent contribution again would not be discretely valued by the original argument of Resta [6], it indeed has this property when treated carefully. We here consider a more general notion of gauge dependence than in the "modern theories" and generalizations thereof, therefore those results are not directly applicable. Nevertheless it may still be the case that the gauge dependent contribution to (24) always evaluates to an element of a set of discrete values and we postpone such an investigation for a later work. Finally, (24) is manifestly invariant under a translation of the energy zero, as one would expect.

## 3.2 Orbital magnetization

We first identify the "atomic-like" contribution [4,16] to the unperturbed magnetization, which arises from the term involving $\boldsymbol{j}_{\beta \boldsymbol{R}';\alpha \boldsymbol{R}''}(\boldsymbol{x}, \boldsymbol{R}; t)$ in (23). We find

$$
\begin{aligned}
\bar{M}^{i(0)} &= \frac{e}{2\Omega_{uc}mc}\mathrm{Re}\sum_{\alpha\beta\boldsymbol{R}'}\left(\epsilon^{iab}\int W_{\beta\boldsymbol{0}}^{*}(\boldsymbol{y})y^{a}\mathfrak{p}^{b}(\boldsymbol{y})W_{\alpha\boldsymbol{R}'-\boldsymbol{R}}(\boldsymbol{y})d\boldsymbol{y}\right)\eta_{\alpha\boldsymbol{R}';\beta\boldsymbol{R}}^{(0)} \\
&= \frac{e}{2\hbar c}\mathrm{Re}\int_{\mathrm{BZ}}\frac{d\boldsymbol{k}}{(2\pi)^{d}}\sum_{n}f_{n\boldsymbol{k}}\epsilon^{iab}\Big((\xi_{nn}^{a}+\mathcal{W}_{nn}^{a})\partial_{b}E_{n\boldsymbol{k}}+i\sum_{m}(E_{m\boldsymbol{k}}-E_{n\boldsymbol{k}})\xi_{nm}^{a}\xi_{mn}^{b} \\
&\qquad\qquad -i\sum_{m}(E_{n\boldsymbol{k}}-E_{m\boldsymbol{k}})\mathcal{W}_{nm}^{a}\xi_{mn}^{b}\Big).
\end{aligned}
\tag{25}
$$

The "itinerant contribution" [4,16], which arises from the term involving $\tilde{\boldsymbol{j}}_{\beta \boldsymbol{R}';\alpha \boldsymbol{R}''}(\boldsymbol{x}, \boldsymbol{R}; t)$ in (23), is found to be

$$
\begin{aligned}
\tilde{M}^{i(0)} &= \frac{e}{2\hbar c}\mathrm{Re}\int_{\mathrm{BZ}}\frac{d\boldsymbol{k}}{(2\pi)^{d}}\sum_{n}f_{n\boldsymbol{k}}\epsilon^{iab}\Big((\xi_{nn}^{a}+\mathcal{W}_{nn}^{a})\partial_{b}E_{n\boldsymbol{k}} \\
&\qquad\qquad +i\sum_{m}(E_{n\boldsymbol{k}}-E_{m\boldsymbol{k}})\mathcal{W}_{nm}^{a}\mathcal{W}_{mn}^{b}+i\sum_{m}(E_{n\boldsymbol{k}}-E_{m\boldsymbol{k}})\mathcal{W}_{nm}^{a}\xi_{mn}^{b}\Big).
\end{aligned}
\tag{26}
$$

In the trivial insulator limit described in Sec. 2, (25) and (26) separately reduce to the usual expressions; taking $f_{n\boldsymbol{k}} \to f_n$ and $\mathcal{W}_{nm}^{a}(\boldsymbol{k}) \neq 0$ only if $f_n = f_m$, and using $i\partial u_{n\boldsymbol{k}}(\boldsymbol{x})/\partial k^{a} = \sum_{m}\xi_{mn}^{a}(\boldsymbol{k})u_{m\boldsymbol{k}}(\boldsymbol{x})$ to find $(\partial_a u_{n\boldsymbol{k}}|H_{\boldsymbol{k}}|\partial_b u_{n\boldsymbol{k}}) = \sum_{m}E_{m\boldsymbol{k}}\xi_{nn}^{a}(\boldsymbol{k})\xi_{mn}^{b}(\boldsymbol{k})$, the expressions for $\boldsymbol{M}_{\mathrm{LC}}$ and $\boldsymbol{M}_{\mathrm{IC}}$ of the "modern theory of magnetization" in a trivial insulator [5] are recovered from (25) and (26), respectively. More generally, combining (25) and (26) we have

$$
\begin{aligned}
M^{i(0)} &= \frac{e}{2\hbar c}\mathrm{Re}\int_{\mathrm{BZ}}\frac{d\boldsymbol{k}}{(2\pi)^{d}}\sum_{n}f_{n\boldsymbol{k}}\epsilon^{iab}\Big(2(\xi_{nn}^{a}+\mathcal{W}_{nn}^{a})\partial_{b}E_{n\boldsymbol{k}} \\
&\qquad\qquad +i\sum_{m}(E_{m\boldsymbol{k}}-E_{n\boldsymbol{k}})(\xi_{nm}^{a}\xi_{mn}^{b}-\mathcal{W}_{nm}^{a}\mathcal{W}_{mn}^{b})\Big) \\
&= \frac{e}{2\hbar c}\mathrm{Re}\int_{\mathrm{BZ}}\frac{d\boldsymbol{k}}{(2\pi)^{d}}\sum_{n}\epsilon^{iab}\Big(2(\partial_{a}f_{n\boldsymbol{k}})(\xi_{nn}^{b}+\mathcal{W}_{nn}^{b})E_{n\boldsymbol{k}}+f_{n\boldsymbol{k}}E_{n\boldsymbol{k}}\partial_{a}\xi_{nn}^{b} \\
&\qquad\qquad +if_{n\boldsymbol{k}}\sum_{m}E_{m\boldsymbol{k}}\xi_{nm}^{a}\xi_{mn}^{b}+iE_{n\boldsymbol{k}}\sum_{m}(f_{m\boldsymbol{k}}-f_{n\boldsymbol{k}})\mathcal{W}_{nm}^{a}\mathcal{W}_{mn}^{b}\Big),
\end{aligned}
\tag{27}
$$

where the second equality follows under the assumption that the integrand is sufficiently well-behaved such that all surface terms can be taken to vanish upon an integration by parts; at first order in the perturbative analysis we also employ such an assumption. Of course, in the trivial insulator limit the usual expression [5] is again recovered; that is, in this limit (27) reduces to

$$
M^{i(0;\mathrm{insulator})} = \frac{e}{2\hbar c}\mathrm{Re}\int_{\mathrm{BZ}}\frac{d\boldsymbol{k}}{(2\pi)^{d}}\sum_{n}\epsilon^{iab}\Big(f_{n\boldsymbol{k}}E_{n\boldsymbol{k}}\partial_{a}\xi_{nn}^{b}+if_{n\boldsymbol{k}}\sum_{m}E_{m\boldsymbol{k}}\xi_{nm}^{a}\xi_{mn}^{b}\Big).
$$

In particular, in that limit (27) is gauge invariant. Moreover, in Appendix C we show that (27) generally vanishes if the unperturbed Hamiltonian is time-reversal symmetric, as expected.

If we again consider the effect of shifting the origin of each ELWF by a Bravais lattice vector $\boldsymbol{R}_{\mathrm{s}}$, we find

$$
M^{i(0)} \to M^{i(0)} + \frac{e}{mc}\epsilon^{iab}R_{\mathrm{s}}^{a}\int_{\mathrm{BZ}}\frac{d\boldsymbol{k}}{(2\pi)^{d}}\sum_{n}f_{n\boldsymbol{k}}\mathfrak{p}_{nn}^{b}(\boldsymbol{k}),
$$

where we have used $\mathfrak{p}_{nn}^{a}(\boldsymbol{k}) = \frac{m}{\hbar}\partial_a E_{n\boldsymbol{k}}$ from (17). The term involving $\boldsymbol{R}_s$ vanishes as the net current that flows in an unperturbed crystal occupying its $T = 0$ ground state is zero; that is,

$$\int_{\text{BZ}} \frac{d\boldsymbol{k}}{(2\pi)^d} \sum_n f_{n\boldsymbol{k}} \mathfrak{p}_{nn}^{a}(\boldsymbol{k}) = 0.$$

Thus, (27) is unaffected by shifting ELWFs, as physically expected. Moreover, it is manifest that (27) is unchanged by a translation of the energy zero.

# 4  First order modifications

We here consider the linearly induced macroscopic charge and current densities, which can be understood to arise from the induced macroscopic polarization and the induced free charge and current densities; in the long-wavelength limit considered here, the induced macroscopic magnetization would be uniform [17] and thus not contribute to (1). Under the frozen-ion approximation that we implement, there are only electronic contributions to such quantities.

## 4.1  Electric polarization

We first consider the contribution to (22) that is first order in $\boldsymbol{E}(\omega)$. Making contact with past work [17, 31], we mention that in general the $\rho_{\beta\boldsymbol{R}';\alpha\boldsymbol{R}''}(\boldsymbol{x}, \boldsymbol{R}; \omega)$ do not involve the electric field and so the only contribution to $\boldsymbol{\mu}_{\boldsymbol{R}}^{(1)}(\omega)$ is "dynamical," arising from the modification of the single-particle density matrix due to $\boldsymbol{E}(\omega)$. Implementing (20) we find

$$P^{i(1)}(\omega) = e^2 E^l(\omega) \int_{\text{BZ}} \frac{d\boldsymbol{k}}{(2\pi)^d} \sum_{mn} \frac{f_{nm,\boldsymbol{k}}\xi_{mn}^l\left(\xi_{nm}^i + \mathcal{W}_{nm}^i\right)}{E_{m\boldsymbol{k}} - E_{n\boldsymbol{k}} - \hbar(\omega + i0^+)}$$
$$+ ie^2 \frac{E^l(\omega)}{\hbar(\omega + i0^+)} \int_{\text{BZ}} \frac{d\boldsymbol{k}}{(2\pi)^d} \sum_n f_{n\boldsymbol{k}} \partial_l\left(\xi_{nn}^i + \mathcal{W}_{nn}^i\right). \tag{28}$$

This expression has two notable features; it is gauge dependent, and it diverges in the dc limit. The gauge dependence is not troubling because induced free charges and currents are also involved here; ultimately it is only the net induced charge and current densities that need be gauge invariant. Also, in the limit of a trivial insulator the second term (that involving $\partial_l(\xi_{nn}^i + \mathcal{W}_{nn}^i)$) vanishes and the expected gauge invariant result is recovered [34]. It is notable however that the distinct terms of (28) are sensitive to different aspects of the gauge transformation; the first term, the interband term, involves only off-diagonal elements of $\mathcal{W}^i(\boldsymbol{k})$, while the second term, the intraband term, involves only diagonal elements. This is to be expected because of the way in which the Lie algebra components of the Berry connection appear. Second, it is notable that a diverging linearly induced polarization in the dc limit is not unprecedented. For example, if one considers a hydrogen atom initially occupying its $2s$ state, dc divergences occur as a result of non-vanishing matrix elements between $2s$ and $2p$ states facilitated by an electric dipole interaction term. Such a divergence could arise from the first term of (28), but does not occur here as we take the crystal to initially occupy its unique electronic ground state, in contrast to this example for the hydrogen atom. So although such a divergence is not entirely novel in principle, the mechanism underlying the divergence of (28) is distinct from that of atomic and molecular physics. We return to this issue in Sec. 5.

## 4.2  Macroscopic bound and free currents

Like the macroscopic polarization and magnetization, the spatial uniformity of the electric field renders the macroscopic bound and free current densities uniform [17]. Thus we do not

indicate any spatial dependence of such quantities. Moreover, both the macroscopic bound and free current densities can found from any one of the "site quantities" used in their construction [16].

Implementing (28) we find the linearly induced macroscopic bound current density [16, 17],

$$J_B^{(1)}(\omega) = -i\omega P^{(1)}(\omega).$$  (29)

This is non-diverging in the $\omega \to 0$ limit, as would be expected physically. Furthermore, in Appendix C we show that (29) vanishes in the $\omega \to 0$ limit if the unperturbed Hamiltonian is time-reversal symmetric.

We now consider the linearly induced macroscopic free current density. The corresponding microscopic density is defined as [16]

$$j_F(x, \omega) \equiv \frac{1}{2}\sum_{RR'} s(x; R, R')I(R, R'; \omega).$$  (30)

From the definitions presented in that past work, we find the first-order modification to the link currents $I(R, R'; \omega)$ to be of the form

$$I^{(1)}(R, R'; \omega) = \frac{e}{i\hbar}\sum_{\alpha\lambda}\left(H_{\alpha R;\lambda R'}^{(1)}(\omega)\eta_{\lambda R';\alpha R}^{(0)} - \eta_{\alpha R;\lambda R'}^{(0)}H_{\lambda R';\alpha R}^{(1)}(\omega)\right)$$
$$+ \frac{e}{i\hbar}\sum_{\alpha\lambda}\left(H_{\alpha R;\lambda R'}^{(0)}\eta_{\lambda R';\alpha R}^{(1)}(\omega) - \eta_{\alpha R;\lambda R'}^{(1)}(\omega)H_{\lambda R';\alpha R}^{(0)}\right).$$  (31)

The first term of (31) is termed a "compositional" modification, arising due to a dependence of the generalized site quantity matrix elements on the electromagnetic field, and the second a "dynamical" modification. An expression for (31) is given in Appendix D, which can explicitly be shown to satisfy

$$I^{(1)}(R, R'; \omega) = -I^{(1)}(R', R; \omega),$$

as required, as well as

$$I^{(1)}(R, R'; \omega) = I^{(1)}(R + R_s, R' + R_s; \omega),$$

and

$$\sum_{R'} I^{(1)}(R, R'; \omega) = 0,$$

as one would physically expect for a translationally invariant system subject to a uniform electric field. The latter can be understood by noting that the electronic "site charges" evolve according to [16]

$$\frac{dQ_R(t)}{dt} = \sum_{R'} I(R, R'; t),$$

and in this case we expect there to be no build up of charge at any particular lattice site; we therefore expect $dQ_R(t)/dt$ to vanish. In fact, from the definition of $Q_R(t)$ and using (20) it can be show that

$$Q_R^{(1)}(\omega) = \sum_{\alpha} \eta_{\alpha R;\alpha R}^{(1)}(\omega) = 0.$$

Then, in Appendix D we show

$$
\begin{aligned}
J_F^{i(1)}(\omega) &= \frac{1}{2\Omega_{uc}} \sum_{\boldsymbol{R}'} (R^i - R'^i) I^{(1)}(\boldsymbol{R}, \boldsymbol{R}'; \omega) \\
&= -\frac{e^2}{\hbar} E^l(\omega) \int_{\text{BZ}} \frac{d\boldsymbol{k}}{(2\pi)^d} \left( \sum_n f_{n\boldsymbol{k}} \partial_i (\xi_{nn}^l + \mathcal{W}_{nn}^l) + \sum_{nm} f_{nm,\boldsymbol{k}} \text{Im}\left[ (\xi_{nm}^l + \mathcal{W}_{nm}^l) \mathcal{W}_{mn}^i \right] \right) \\
&\quad + \frac{ie^2}{\hbar} E^l(\omega) \int_{\text{BZ}} \frac{d\boldsymbol{k}}{(2\pi)^d} \sum_{nm} f_{nm,\boldsymbol{k}} \xi_{mn}^l \mathcal{W}_{nm}^i \left( 1 + \frac{\hbar\omega}{E_{m\boldsymbol{k}} - E_{n\boldsymbol{k}} - \hbar(\omega + i0^+)} \right) \\
&\quad + \frac{ie^2}{\hbar} \frac{E^l(\omega)}{\hbar(\omega + i0^+)} \int_{\text{BZ}} \frac{d\boldsymbol{k}}{(2\pi)^d} \sum_n f_{n\boldsymbol{k}} \partial_l \partial_i E_{n\boldsymbol{k}} ,
\end{aligned}
\tag{32}
$$

where the first term in the second equality results from the compositional modification of (31), while the second and third terms from the dynamical modification. Notably $J_F^{(1)}(\omega)$ diverges in the dc limit, which is as one would physically expect given that we have not accounted for any scattering mechanisms. In fact, it is the third term in the second equality of (32) that will lead to the dc divergence of the electrical conductivity tensor; this term involves the second term of (20). Moreover, $J_F^{(1)}(\omega)$ is gauge dependent, akin to $J_B^{(1)}(\omega)$, and it is only through this gauge dependence that (32) involves "interband" contributions. Notably if all of the energy bands of the unperturbed crystal were isolated from one another and the corresponding Hilbert bundles assumed trivial, then one can take $U_{n\alpha}(\boldsymbol{k})$ proportional to $\delta_{n\alpha}$ such that $\mathcal{W}_{nm}^a(\boldsymbol{k})$ is proportional to $\delta_{nm}$ and thus the interband contributions to $J_F^{(1)}(\omega)$ vanish; in this limiting case the induced free current density involves only "intraband" contributions, which is as one would expect for simple models.

## 4.3 Time-reversal symmetry

The general expression (28) that we derive for $P^{(1)}(\omega)$ has the feature that it contains an intraband contribution and this contribution diverges in the dc limit. From the simple picture of polarization presented in Sec. 1, the presence of such a contribution is unexpected. While in general our description thus asserts that this simple picture is not complete, in Appendix C we show that such an intraband contribution vanishes if the unperturbed Hamiltonian is time-reversal symmetric and $P^{(1)}(\omega)$ takes the more expected form

$$
P^{i(1)}(\omega) \overset{\mathcal{T}}{=} e^2 E^l(\omega) \int_{\text{BZ}} \frac{d\boldsymbol{k}}{(2\pi)^d} \sum_{mn} \frac{f_{nm,\boldsymbol{k}} \xi_{mn}^l (\xi_{nm}^i + \mathcal{W}_{nm}^i)}{E_{m\boldsymbol{k}} - E_{n\boldsymbol{k}} - \hbar(\omega + i0^+)} .
$$

Here $\overset{\mathcal{T}}{=}$ will denote an equality that holds in the presence of time-reversal symmetry. Adopting the approach of (3), we find

$$
\epsilon_{\text{inter}}^{il}(\omega) \overset{\mathcal{T}}{=} 4\pi e^2 \int_{\text{BZ}} \frac{d\boldsymbol{k}}{(2\pi)^d} \sum_{mn} \frac{f_{nm,\boldsymbol{k}} \xi_{mn}^l (\xi_{nm}^i + \mathcal{W}_{nm}^i)}{E_{m\boldsymbol{k}} - E_{n\boldsymbol{k}} - \hbar(\omega + i0^+)} ,
$$

which, apart from the gauge dependence, is consistent with the insight from analogies with molecular response and the more simple approaches mentioned in Sec. 1. That is, for crystalline solids in which time-reversal symmetry holds, it is only interband contributions that are involved in $P^{(1)}(\omega)$. However, even in this simple case $P^{(1)}(\omega)$ and $\epsilon_{\text{inter}}^{il}(\omega)$ remain gauge dependent, and thus the introduction of ELWFs and the ambiguity in their choice need be involved in any discussion of such quantities. In this case the induced macroscopic free current

density (32) reduces to

$$
J_F^{i(1)}(\omega) \stackrel{\mathcal{T}}{=} ie^2 \omega E^l(\omega) \int_{\text{BZ}} \frac{d\boldsymbol{k}}{(2\pi)^d} \sum_{nm} \frac{f_{nm,\boldsymbol{k}} \xi_{mn}^l \mathcal{W}_{nm}^i}{E_{m\boldsymbol{k}} - E_{n\boldsymbol{k}} - \hbar(\omega + i0^+)}
$$
$$
+ \frac{ie^2}{\hbar} \frac{E^l(\omega)}{\hbar(\omega + i0^+)} \int_{\text{BZ}} \frac{d\boldsymbol{k}}{(2\pi)^d} \sum_n f_{n\boldsymbol{k}} \partial_l \partial_i E_{n\boldsymbol{k}} \,,
$$

still having both interband and intraband contributions. Notably the interband contribution is gauge dependent and cancels with the gauge dependent term appearing in the induced bound current density $-i\omega \boldsymbol{P}^{(1)}(\omega)$ and thus the net induced current density is gauge independent, as one would expect. In the special case of isolated bands, $U_{n\alpha}(\boldsymbol{k})$ can be chosen proportional to $\delta_{n\alpha}$ and the gauge dependent contributions to $\boldsymbol{P}^{(1)}(\omega)$ and $\boldsymbol{J}_F^{(1)}(\omega)$ separately vanish. In addition, if the "parabolic band approximation" is implemented, that is, if one takes each energy eigenvalue of an occupied state to be $E_{n\boldsymbol{k}} = \hbar^2 |\boldsymbol{k}|^2 / 2m$, $\boldsymbol{J}_F^{(1)}(\omega)$ agrees with (2) after the identification $m_0 = m$ and $N = N_{\text{el}}$. In fact, we find under the parabolic band approximation that

$$
\epsilon_{\text{eff}}^{il}(\omega) \stackrel{\mathcal{T}}{=} \delta^{il} + 4\pi e^2 \int_{\text{BZ}} \frac{d\boldsymbol{k}}{(2\pi)^d} \sum_{mn} \frac{f_{nm,\boldsymbol{k}} \xi_{mn}^l \xi_{nm}^i}{E_{m\boldsymbol{k}} - E_{n\boldsymbol{k}} - \hbar(\omega + i0^+)} - \frac{4\pi e^2 N_{\text{el}}}{m\omega(\omega + i0^+)} \delta^{il} \,,
$$

or

$$
\sigma^{il}(\omega) \stackrel{\mathcal{T}}{=} -ie^2 \hbar \omega \int_{\text{BZ}} \frac{d\boldsymbol{k}}{(2\pi)^d} \sum_{mn} \frac{f_{nm,\boldsymbol{k}} \xi_{mn}^l \xi_{nm}^i}{E_{m\boldsymbol{k}} - E_{n\boldsymbol{k}} - \hbar(\omega + i0^+)} + \frac{ie^2 N_{\text{el}}}{m(\omega + i0^+)} \delta^{il}
$$
$$
\stackrel{\mathcal{T}}{=} ie^2 \int_{\text{BZ}} \frac{d\boldsymbol{k}}{(2\pi)^d} \sum_{mn} \frac{f_{nm,\boldsymbol{k}}(E_{n\boldsymbol{k}} - E_{m\boldsymbol{k}}) \xi_{mn}^l \xi_{nm}^i}{E_{m\boldsymbol{k}} - E_{n\boldsymbol{k}} - \hbar(\omega + i0^+)} + \frac{ie^2 N_{\text{el}}}{m(\omega + i0^+)} \delta^{il} \,.
$$

In moving from the first to the second equality in this expression for $\sigma^{il}(\omega)$, relations that hold only in the presence of time-reversal symmetry are implemented. However, we will find that this latter form of $\sigma^{il}(\omega)$ holds even in the absence of time-reversal symmetry. Moreover, in the absence of that symmetry $\boldsymbol{P}^{(1)}(\omega)$ takes the more complicated form (28), which generally involves intraband contributions. This results in $\boldsymbol{P}^{(1)}(\omega)$ having a more general gauge dependence and as well $\boldsymbol{J}_F^{(1)}(\omega)$ having a more general gauge dependence. However, as was the case here, when these more general expressions are combined, for instance when constructing $\epsilon_{\text{eff}}^{il}(\omega)$ or $\sigma^{il}(\omega)$, the gauge dependent terms again cancel.

## 4.4  Induced macroscopic current density

Returning to the more general investigation, and thus allowing the possible breaking of time-reversal symmetry, we again find that although (29) and (32) are not individually gauge invariant and thus are not themselves directly physically observable, their sum is. Indeed, combining (29) and (32) we find

$$
J^{i(1)}(\omega) = \frac{ie^2}{\hbar} E^l(\omega) \int_{\text{BZ}} \frac{d\boldsymbol{k}}{(2\pi)^d} \sum_{mn} \frac{f_{nm,\boldsymbol{k}}(E_{n\boldsymbol{k}} - E_{m\boldsymbol{k}}) \xi_{mn}^l \xi_{nm}^i}{E_{m\boldsymbol{k}} - E_{n\boldsymbol{k}} - \hbar(\omega + i0^+)}
$$
$$
+ \frac{ie^2}{\hbar} \frac{E^l(\omega)}{\hbar(\omega + i0^+)} \int_{\text{BZ}} \frac{d\boldsymbol{k}}{(2\pi)^d} \sum_n f_{n\boldsymbol{k}} \partial_l \partial_i E_{n\boldsymbol{k}} \,.
$$

The first term comes from taking $-i\omega = (E_{n\boldsymbol{k}} - E_{m\boldsymbol{k}}) + (E_{m\boldsymbol{k}} - E_{n\boldsymbol{k}} - i\omega)$ in the interband contribution of $\boldsymbol{P}^{(1)}(\omega)$ to $\boldsymbol{J}_B^{(1)}(\omega)$ ((28) to (29)). Then it is only the term that is gauge invariant and explicitly energy dependent in this particular contribution to $\boldsymbol{J}_B^{(1)}(\omega)$ that is not

cancelled when combined with $J_F^{(1)}(\omega)$, (32). In particular, it is part of the first term in the second equality of (32) (all but that involving $f_{nm,\boldsymbol{k}}\text{Im}[\xi_{nm}^l \mathcal{W}_{mn}^i]$), which is a compositional modification, that combines with the second term of the contribution of (28) to (29) when we calculate $\boldsymbol{J}^{(1)}(\omega)$, and ultimately it is the combination of these terms that cancel with the interband contribution of (28) to (29) that is gauge invariant and does not explicitly depend on energy. The remaining gauge dependent terms all involve products of the form $f_{nm,\boldsymbol{k}}\xi_{nm}^a \mathcal{W}_{mn}^b$ and cancel one another. The second term arises from the induced free current density alone and is the only term in (32) that does not cancel with terms from (29). While the "origin" of each of the terms can most easily be seen in the above form of the expression, it can be rewritten in the more familiar form

$$
\begin{aligned}
J^{i(1)}(\omega) = &-ie^2\omega E^l(\omega)\int_{\text{BZ}}\frac{d\boldsymbol{k}}{(2\pi)^d}\sum_{mn}f_{n\boldsymbol{k}}\frac{E_{m\boldsymbol{k}}-E_{n\boldsymbol{k}}}{(E_{m\boldsymbol{k}}-E_{n\boldsymbol{k}})^2-(\hbar(\omega+i0^+))^2}\left(\xi_{nm}^i\xi_{mn}^l+\xi_{nm}^l\xi_{mn}^i\right)\\
&-\frac{ie^2}{\hbar}E^l(\omega)\int_{\text{BZ}}\frac{d\boldsymbol{k}}{(2\pi)^d}\sum_{mn}f_{n\boldsymbol{k}}\frac{(E_{m\boldsymbol{k}}-E_{n\boldsymbol{k}})^2}{(E_{m\boldsymbol{k}}-E_{n\boldsymbol{k}})^2-(\hbar(\omega+i0^+))^2}\left(\xi_{nm}^i\xi_{mn}^l-\xi_{nm}^l\xi_{mn}^i\right)\\
&+\frac{ie^2}{\hbar}\frac{E^l(\omega)}{\hbar(\omega+i0^+)}\int_{\text{BZ}}\frac{d\boldsymbol{k}}{(2\pi)^d}\sum_{n}f_{n\boldsymbol{k}}\partial_l\partial_i E_{n\boldsymbol{k}}.
\end{aligned}
\tag{33}
$$

This is in agreement with usual perturbative calculations that implement the minimal coupling Hamiltonian. In particular, using (17) to rewrite the integrands of (33) to involve velocity matrix elements $\mathfrak{v}_{nn'}(\boldsymbol{k})=\mathfrak{p}_{nn'}(\boldsymbol{k})/m$, for example, Eq. (25,26) of Allen [35] are reproduced.

The final term of (33) can be understood as a "Drude" contribution. This term follows from the final term of (20), and enters here via the induced free current density (32). Notably, such a term can lead to an induced current density that is orthogonal to the applied electric field. This is not to be confused with the well-understood anomalous Hall conductivity however, because in this case since the Cartesian components $i$ and $l$ are symmetric there exists a basis in which this contribution to the conductivity tensor is diagonal. Physically this means that, were the applied electric field characterized by a single non-vanishing component with respect to such a basis, the induced current density arising from this term would be parallel to that field. Thus, we understand the possibility of such an induced orthogonal current density to be entirely a consequence of crystalline anisotropy.

In contrast, the first and second terms of (33) are related to both the induced bound and free current densities. Notably, the second term can be understood as a finite-frequency generalization of the "anomalous Hall" current density [36]. This portion of the induced current density is unique because, unlike the contribution from final term of (33), the spatial components $i$ and $l$ are asymmetric and consequently there does *not* exist a basis in which this contribution is diagonal; there does not exist a basis in which the induced current associated with this term is parallel to the applied electric field.

## 4.5   Microscopic charge and current densities

The divergence of (28) in the dc limit may raise concerns about our identification of the polarization. We are thus motivated to consider the first-order modifications of the expectation values of the electronic charge and current density operators due to $\boldsymbol{E}(\omega)$ – quantities that could be found from traditional perturbation theory with the minimal coupling Hamiltonian[16] – with the hope that further insight might be gained. Implementing (20) into previously de-

---

[16]See, e.g., Chapter 6 of [45].

veloped expressions [16], we find

$$\langle \hat{\rho}(\boldsymbol{x}, \omega) \rangle^{(1)} = e^2 E^l(\omega) \int_{\text{BZ}} \frac{d\boldsymbol{k}}{(2\pi)^d} \sum_{mn} \frac{f_{nm,\boldsymbol{k}} \xi^l_{mn}}{E_{m\boldsymbol{k}} - E_{n\boldsymbol{k}} - \hbar(\omega + i0^+)} \psi^*_{n\boldsymbol{k}}(\boldsymbol{x}) \psi_{m\boldsymbol{k}}(\boldsymbol{x})$$
$$+ ie^2 \frac{E^l(\omega)}{\hbar(\omega + i0^+)} \int_{\text{BZ}} \frac{d\boldsymbol{k}}{(2\pi)^d} \sum_n f_{n\boldsymbol{k}} \frac{\partial}{\partial k^l} \big( \psi^*_{n\boldsymbol{k}}(\boldsymbol{x}) \psi_{n\boldsymbol{k}}(\boldsymbol{x}) \big), \tag{34}$$

and

$$\langle \hat{j}^i(\boldsymbol{x}, \omega) \rangle^{(1)} = \frac{e^2}{m} E^l(\omega) \int_{\text{BZ}} \frac{d\boldsymbol{k}}{(2\pi)^d} \sum_{mn} \frac{f_{nm,\boldsymbol{k}} \xi^l_{mn}}{E_{m\boldsymbol{k}} - E_{n\boldsymbol{k}} - \hbar(\omega + i0^+)} \psi^*_{n\boldsymbol{k}}(\boldsymbol{x}) \mathfrak{p}^i(\boldsymbol{x}) \psi_{m\boldsymbol{k}}(\boldsymbol{x})$$
$$+ \frac{ie^2}{m} \frac{E^l(\omega)}{\hbar(\omega + i0^+)} \int_{\text{BZ}} \frac{d\boldsymbol{k}}{(2\pi)^d} \sum_n f_{n\boldsymbol{k}} \frac{\partial}{\partial k^l} \big( \psi^*_{n\boldsymbol{k}}(\boldsymbol{x}) \mathfrak{p}^i(\boldsymbol{x}) \psi_{n\boldsymbol{k}}(\boldsymbol{x}) \big). \tag{35}$$

The electronic charge and current density operators that we implement are those that arise via Noether's theorem and thus satisfy the continuity equation

$$\frac{\partial}{\partial t} \hat{\rho}(\boldsymbol{x}, t) + \frac{\partial}{\partial x^a} \hat{j}^a(\boldsymbol{x}, t) = 0.$$

Assuming an expansion of these operators in powers of the electric field exists, continuity must then hold at each order in $\boldsymbol{E}(\omega)$. The same must then be true of the expectation values of such operators. This can explicitly be shown to be the case at first order in $\boldsymbol{E}(\omega)$; implementing (34,35), we find

$$-i\omega \langle \hat{\rho}(\boldsymbol{x}, \omega) \rangle^{(1)} + \frac{\partial}{\partial x^a} \langle \hat{j}^a(\boldsymbol{x}, \omega) \rangle^{(1)} = 0,$$

given that in principle charge continuity holds in the unperturbed system in a perturbative scheme.

Notably, (34) has a dc divergence taking a form similar to that of (28). Like that second term of (28), the second term of (34) vanishes if the unperturbed system is time-reversal symmetric, although this symmetry does not cause the second term of (35) to vanish. Thus, it appears that if one insists on defining electric multipole moments by way of partitioning the electronic charge density into portions that are used to define "site" polarization fields from which "site" multipole moments are extracted and summed to give the full electric multipole moments of the crystal, whether that be via the approach we implement here or some other method, it is unavoidable that one will find a such a dc divergence. In a sense, this unexpected dc divergence is not arising as a consequence of our identification of the polarization, but rather it is inherent to the induced charge density at low frequencies.

## 5 Conclusion

In this work we have considered how polarization and magnetization fields can be defined for metallic systems. In contrast to the approach of the "modern theories of polarization and magnetization," we employ a previously developed strategy [16] for defining microscopic polarization and magnetization fields in general crystalline solids, the macroscopic analogues of which are defined by spatial averaging. Exponentially localized Wannier functions play a central role in how the electronic components of such quantities are defined. In a trivial insulator the macroscopic charge and current densities can be obtained from the macroscopic polarization and magnetization fields alone, both for the ground state and in linear response,

while for a metal one would naturally expect contributions from the macroscopic free charge and free current densities, and we have identified them here.

We implemented this approach for a simple instance of a metal, a $p$-doped semiconductor, initially occupying its $T = 0$ ground state, and we assume that the Hilbert bundle over the first Brillouin zone associated with any set of isolated energy bands is globally trivial. With this, and because we assume the existence of a band gap above the Fermi energy, contact with expressions for a trivial insulator can readily be reached as a limiting case of the more general expressions we obtain. Indeed, in Sec. 3 we employ the general definitions in this setting to obtain expressions for $P^{(0)}$ and $M^{(0)}$, and in the limit of vanishing doping our expressions reduced to those of the "modern theories." While in that limit $P^{(0)}$ is unique modulo a "quantum of ambiguity" and $M^{(0)}$ is gauge-invariant, this is not so for a metal. Nonetheless, $P^{(0)}$ exhibits the expected property that under translation of the origin of all ELWFs by a Bravais lattice vector $R$, $P^{(0)}$ is changed by an additive constant proportional to $R$; $M^{(0)}$ is unaffected by such a translation, and both quantities are unchanged by a shift of the energy zero.

Although the expressions we obtain for $P^{(0)}$ and $M^{(0)}$ agree with the "modern theories" in the limit of a trivial insulator, the two approaches disagree more generally. In the "modern theory of polarization" it has been argued that $P^{(0)}$ is not well-defined in metallic systems [9,11]. In the approach implemented here, a definition is always admitted and we obtain a $P^{(0)}$ that is formally similar to that of a trivial insulator. Meanwhile, the "modern theory of magnetization" has been generalized using thermodynamic arguments to obtain an expression for $M^{(0)}$ valid for metals and Chern insulators [11,12], but even so the expression we derive does not agree. This disagreement is not surprising; there is an inherent ambiguity in what one might identify as a magnetization, and the underlying philosophies of these approaches differ. We consider polarization and magnetization to fundamentally arise as microscopic quantities from which macroscopic analogues are obtained, while the "modern theories" view such quantities as being fundamentally macroscopic. These differences are elucidated in the way the expressions for $M^{(0)}$ differ; we find $M^{(0)}$ to be gauge dependent, owing to the central role played by a set of ELWFs in its identification, while in the "modern theory" it is found to explicitly involve a chemical potential, even in the case of a Chern insulator, emphasizing the inherent thermodynamic considerations and the assumed relation to finite-sized systems. In bulk crystals both approaches are valid, each with positive features particularly evident in the domain of considerations that motivate them. Some advantages of the approach implemented here is that the polarization and magnetization are on the same footing, both being defined for all media, that definitions for free charge and current densities are admitted, and that the charge and current densities (1) arise directly from an analysis of the underlying microscopic theory.

In Sec. 4 we investigated the linear response of a metallic crystal to an optical field at finite frequency $\omega$, a more general response than is typically considered in the "modern theories." We considered the "long-wavelength limit," within the independent particle and frozen-ion approximations, where the applied electric field is taken to be the macroscopic Maxwell field. Here only $P(t)$ and $J_F(t)$ make a contribution to the linearly induced macroscopic current density, $J^{(1)}(t) = \partial P^{(1)}(t)/\partial t + J_F^{(1)}(t)$. While in elementary models of the optical response of metals $\partial P^{(1)}(t)/\partial t$ is associated with interband response and $J_F^{(1)}(t)$ with intraband response, here we find a more general scenario; in general, that simple association is no longer the case and both contributions are gauge dependent. However, we do find that if all of the energy bands of the unperturbed crystal were isolated from one another then $J_F^{(1)}(t)$ would have only intraband contributions and would be gauge invariant, in agreement with those more simple models. Nevertheless, the general $\sigma^{il}(\omega)$ we obtain is gauge invariant and reproduces the usual conductivity tensor of a metal, consisting of a finite-frequency generalization of the "anomalous Hall" and a "Drude" contribution; the latter is entirely due to $J_F^{(1)}(\omega)$.

We also found that if an unperturbed metallic crystal violates time-reversal symmetry, then

there is a term in the linear response of the microscopic charge density proportional to $\omega^{-1}$; in an approach such as ours that relates the macroscopic polarization to electric dipole moments associated with "site" contributions to the microscopic charge density, this leads to a term in $\boldsymbol{P}^{(1)}(\omega)$ proportional to $\omega^{-1}$. It is the same mechanism that gives rise to the dc divergences of both $\boldsymbol{P}^{(1)}(\omega)$ and $\boldsymbol{J}_F^{(1)}(\omega)$. That is, both divergences involve the second term of (20), which we show in Appendix B is a consequence of an interaction term that gives rise to the intraband response; in this way of identifying inter- and intraband contributions to the linear response we can make contact with earlier work by Blount and others, although the formalism in which we work is indeed much more general.

Given that the association of $\boldsymbol{J}_F^{(1)}(t)$ with intraband response and $\partial \boldsymbol{P}^{(1)}(t)/\partial t$ with interband response does not hold, that both contributions are gauge dependent, and that $\boldsymbol{P}^{(1)}(\omega)$ involves a term proportional to $\omega^{-1}$, one could argue that a different definition of polarization would be more appropriate. However, such a purported new polarization could not be associated with the dipole moment of microscopic charge densities localized about individual lattice sites. In fact, a more general argument could be made against the philosophy of our investigations. Our goal, a critic might assert, should be to seek what could be taken as "unique" definitions of $\boldsymbol{P}$, $\boldsymbol{M}$, $\varrho_F$, and $\boldsymbol{J}_F$, and for a metal we do not even demonstrate that for $\boldsymbol{P}$ and $\boldsymbol{M}$ in the ground state. We would reply that such uniqueness is not a reasonable goal. After all, even in the ground state of a trivial insulator the value of $\boldsymbol{P}$ is subject to a "quantum of ambiguity." And once one moves to a general temporal and spatial dependence there are clearly a host of fields $\boldsymbol{P}(\boldsymbol{x},t)$, $\boldsymbol{M}(\boldsymbol{x},t)$, $\varrho_F(\boldsymbol{x},t)$, and $\boldsymbol{J}_F(\boldsymbol{x},t)$ that could be used to describe the physical quantities $\varrho(\boldsymbol{x},t)$ and $\boldsymbol{J}(\boldsymbol{x},t)$ via (1). Our perspective is that the focus should be on exploring what might be useful ways of introducing such quantities, for the purpose of both physical insight and calculation. Within that framework this paper can be taken as one such contribution.

## Acknowledgements

We thank Jason Kattan for useful discussions. This work was supported by the Natural Sciences and Engineering Research Council of Canada (NSERC). P. T. M. acknowledges an Ontario Graduate Scholarship.

## A  Perturbation theory – A strict approach

Recall Eq. (37) of [16], the general equations of motion for the single-particle density matrix $\eta_{\alpha\boldsymbol{R};\beta\boldsymbol{R'}}$. We here consider the long-wavelength limit, $\boldsymbol{E}(\boldsymbol{x},t) \to \boldsymbol{E}(t)$ and $\boldsymbol{B}(\boldsymbol{x},t) \to \boldsymbol{0}$, of those general expressions. This yields

$$i\hbar \frac{\partial}{\partial t}\eta_{\alpha\boldsymbol{R};\beta\boldsymbol{R'}}(t) = \sum_{\lambda\boldsymbol{R''}}\left(\bar{H}_{\alpha\boldsymbol{R};\lambda\boldsymbol{R''}}(t)\eta_{\lambda\boldsymbol{R''};\beta\boldsymbol{R'}}(t) - \eta_{\alpha\boldsymbol{R};\lambda\boldsymbol{R''}}(t)\bar{H}_{\lambda\boldsymbol{R''};\beta\boldsymbol{R'}}(t)\right)$$
$$- e\Omega_{\boldsymbol{R'}}^0(\boldsymbol{R};t)\eta_{\alpha\boldsymbol{R};\beta\boldsymbol{R'}}(t), \tag{A.1}$$

where, in this limit, $\Omega_y^0(\boldsymbol{x}, t) = \boldsymbol{E}(t) \cdot (\boldsymbol{x} - \boldsymbol{y})$ and

$$
\begin{aligned}
\bar{H}_{\alpha\boldsymbol{R};\beta\boldsymbol{R}'}(t) &= \int W_{\alpha\boldsymbol{R}}^*(\boldsymbol{x}) H_0(\boldsymbol{x}, \mathfrak{p}(\boldsymbol{x})) W_{\beta\boldsymbol{R}'}(\boldsymbol{x}) d\boldsymbol{x} \\
&\quad - \frac{e}{2} \int W_{\alpha\boldsymbol{R}}^*(\boldsymbol{x}) \big(\Omega_{\boldsymbol{R}}^0(\boldsymbol{x}; t) + \Omega_{\boldsymbol{R}'}^0(\boldsymbol{x}; t)\big) W_{\beta\boldsymbol{R}'}(\boldsymbol{x}) d\boldsymbol{x} \\
&= \bar{H}_{\alpha\boldsymbol{R};\beta\boldsymbol{R}'}^{(0)} - \frac{e}{2} E^l(t) \int W_{\alpha\boldsymbol{R}}^*(\boldsymbol{x}) (x^l - R^l) W_{\beta\boldsymbol{R}'}(\boldsymbol{x}) d\boldsymbol{x} \\
&\quad - \frac{e}{2} E^l(t) \int W_{\alpha\boldsymbol{R}}^*(\boldsymbol{x}) (x^l - R'^l) W_{\beta\boldsymbol{R}'}(\boldsymbol{x}) d\boldsymbol{x}.
\end{aligned}
$$

Assuming valid power series expansions for all quantities with respect to the applied electric field $\boldsymbol{E}(t)$, we have

$$
\begin{aligned}
&i\hbar \frac{\partial}{\partial t} \big(\eta_{\alpha\boldsymbol{R};\beta\boldsymbol{R}'}^{(0)}(t) + \eta_{\alpha\boldsymbol{R};\beta\boldsymbol{R}'}^{(1)}(t) + \dots\big) \\
&= \sum_{\lambda\boldsymbol{R}''} \bar{H}_{\alpha\boldsymbol{R};\lambda\boldsymbol{R}''}^{(0)} \big(\eta_{\lambda\boldsymbol{R}'';\beta\boldsymbol{R}'}^{(0)}(t) + \eta_{\lambda\boldsymbol{R}'';\beta\boldsymbol{R}'}^{(1)}(t) + \dots\big) - \sum_{\lambda\boldsymbol{R}''} \big(\eta_{\alpha\boldsymbol{R};\lambda\boldsymbol{R}''}^{(0)}(t) + \eta_{\alpha\boldsymbol{R};\lambda\boldsymbol{R}''}^{(1)}(t) + \dots\big) \bar{H}_{\lambda\boldsymbol{R}'';\beta\boldsymbol{R}'}^{(0)} \\
&\quad - eE^l(t) \sum_{\lambda\boldsymbol{R}''} \Big(\int W_{\alpha\boldsymbol{R}-\boldsymbol{R}''}^*(\boldsymbol{x}) x^l W_{\lambda\boldsymbol{0}}(\boldsymbol{x}) d\boldsymbol{x}\Big) \big(\eta_{\lambda\boldsymbol{R}'';\beta\boldsymbol{R}'}^{(0)}(t) + \eta_{\lambda\boldsymbol{R}'';\beta\boldsymbol{R}'}^{(1)}(t) + \dots\big) \\
&\quad + eE^l(t) \sum_{\lambda\boldsymbol{R}''} \big(\eta_{\alpha\boldsymbol{R};\lambda\boldsymbol{R}''}^{(0)}(t) + \eta_{\alpha\boldsymbol{R};\lambda\boldsymbol{R}''}^{(1)}(t) + \dots\big) \Big(\int W_{\lambda\boldsymbol{R}''-\boldsymbol{R}'}^*(\boldsymbol{x}) x^l W_{\beta\boldsymbol{0}}(\boldsymbol{x}) d\boldsymbol{x}\Big) \\
&\quad - eE^l(t)(R^l - R'^l)\big(\eta_{\alpha\boldsymbol{R};\beta\boldsymbol{R}'}^{(0)}(t) + \eta_{\alpha\boldsymbol{R};\beta\boldsymbol{R}'}^{(1)}(t) + \dots\big),
\end{aligned}
$$

where we have used the translation property $W_{\alpha\boldsymbol{R}}(\boldsymbol{x} - \boldsymbol{R}_1) = W_{\alpha\boldsymbol{R}+\boldsymbol{R}_1}(\boldsymbol{x})$ of the ELWFs. Upon "matching powers" of $\boldsymbol{E}(t)$ on the LHS and RHS, the above is equally expressed as a collection of independent equations,

$$
i\hbar \frac{\partial}{\partial t} \eta_{\alpha\boldsymbol{R};\beta\boldsymbol{R}'}^{(0)}(t) = \sum_{\lambda\boldsymbol{R}''} \Big(\bar{H}_{\alpha\boldsymbol{R};\lambda\boldsymbol{R}''}^{(0)} \eta_{\lambda\boldsymbol{R}'';\beta\boldsymbol{R}'}^{(0)}(t) - \eta_{\alpha\boldsymbol{R};\lambda\boldsymbol{R}''}^{(0)}(t) \bar{H}_{\lambda\boldsymbol{R}'';\beta\boldsymbol{R}'}^{(0)}\Big), \tag{A.2}
$$

$$
\begin{aligned}
i\hbar \frac{\partial}{\partial t} \eta_{\alpha\boldsymbol{R};\beta\boldsymbol{R}'}^{(1)}(t) &= \sum_{\lambda\boldsymbol{R}''} \Big(\bar{H}_{\alpha\boldsymbol{R};\lambda\boldsymbol{R}''}^{(0)} \eta_{\lambda\boldsymbol{R}'';\beta\boldsymbol{R}'}^{(1)}(t) - \eta_{\alpha\boldsymbol{R};\lambda\boldsymbol{R}''}^{(1)}(t) \bar{H}_{\lambda\boldsymbol{R}'';\beta\boldsymbol{R}'}^{(0)}\Big) \\
&\quad - eE^l(t) \sum_{\lambda\boldsymbol{R}''} \Big(\int W_{\alpha\boldsymbol{R}-\boldsymbol{R}''}^*(\boldsymbol{x}) x^l W_{\lambda\boldsymbol{0}}(\boldsymbol{x}) d\boldsymbol{x}\Big) \eta_{\lambda\boldsymbol{R}'';\beta\boldsymbol{R}'}^{(0)}(t) \\
&\quad + eE^l(t) \sum_{\lambda\boldsymbol{R}''} \eta_{\alpha\boldsymbol{R};\lambda\boldsymbol{R}''}^{(0)}(t) \Big(\int W_{\lambda\boldsymbol{R}''-\boldsymbol{R}'}^*(\boldsymbol{x}) x^l W_{\beta\boldsymbol{0}}(\boldsymbol{x}) d\boldsymbol{x}\Big) \\
&\quad - eE^l(t)(R^l - R'^l) \eta_{\alpha\boldsymbol{R};\beta\boldsymbol{R}'}^{(0)}(t) \\
&\equiv \sum_{\lambda\boldsymbol{R}''} \Big(\bar{H}_{\alpha\boldsymbol{R};\lambda\boldsymbol{R}''}^{(0)} \eta_{\lambda\boldsymbol{R}'';\beta\boldsymbol{R}'}^{(1)}(t) - \eta_{\alpha\boldsymbol{R};\lambda\boldsymbol{R}''}^{(1)}(t) \bar{H}_{\lambda\boldsymbol{R}'';\beta\boldsymbol{R}'}^{(0)}\Big) + Q_{\alpha\boldsymbol{R};\beta\boldsymbol{R}'}^{(1)}(t), \tag{A.3}
\end{aligned}
$$

etc.

From (A.2) we recognize that $\eta_{\alpha\boldsymbol{R};\beta\boldsymbol{R}'}^{(0)}(t)$ evolves as the unperturbed single-particle density matrix $\langle \text{gs}| e^{i\hat{H}_0 t/\hbar} \hat{a}_{\alpha\boldsymbol{R}}^\dagger \hat{a}_{\beta\boldsymbol{R}'} e^{-i\hat{H}_0 t/\hbar} |\text{gs}\rangle$ under the unperturbed Hamiltonian $\hat{H}_0$, as we expect. In particular, starting from the equation of motion for the unperturbed electron Green function $i\langle \text{gs}| \hat{\psi}_0^\dagger(\boldsymbol{y}, t) \hat{\psi}_0(\boldsymbol{x}, t) |\text{gs}\rangle$, the related single-particle density matrix evolves as (A.2); the argument is analogous to that which yields (A.1) from the "global" Green function (which is related to the minimal coupling Green function by a generalized Peierls phase). Now, via (8,9)

the relation between the operators generating ELWFs and those generating the $|\psi_{n\boldsymbol{k}}\rangle$ is found to be

$$\hat{a}^{\dagger}_{\alpha\boldsymbol{R}} = \sqrt{\frac{\Omega_{uc}}{(2\pi)^d}} \int_{\text{BZ}} d\boldsymbol{k}\, e^{-i\boldsymbol{k}\cdot\boldsymbol{R}} \sum_n U_{n\alpha}(\boldsymbol{k}) \hat{a}^{\dagger}_{n\boldsymbol{k}}, \tag{A.4}$$

which we then implement to find

$$\eta^{(0)}_{\alpha\boldsymbol{R};\beta\boldsymbol{R}'} = \Omega_{uc} \int_{\text{BZ}} \frac{d\boldsymbol{k}}{(2\pi)^d} e^{i\boldsymbol{k}\cdot(\boldsymbol{R}-\boldsymbol{R}')} \sum_n f_{n\boldsymbol{k}} U^{\dagger}_{\alpha n}(\boldsymbol{k}) U_{n\beta}(\boldsymbol{k}), \tag{A.5}$$

which is independent of time.

We now consider (A.3) and will closely follow the procedure of Appendix B of [31], however we will not introduce filling factors associated with the ELWFs. It is useful to define the intermediate quantity

$$\eta_{m\boldsymbol{k};n\boldsymbol{k}'}(t) \equiv \sum_{\mu\nu\boldsymbol{R}_1\boldsymbol{R}_2} \langle\psi_{m\boldsymbol{k}}|\mu\boldsymbol{R}_1\rangle \eta_{\mu\boldsymbol{R}_1;\nu\boldsymbol{R}_2}(t) \langle\nu\boldsymbol{R}_2|\psi_{n\boldsymbol{k}'}\rangle, \tag{A.6}$$

and from (A.3) it follows that

$$i\hbar\frac{\partial\eta^{(1)}_{m\boldsymbol{k};n\boldsymbol{k}'}(t)}{\partial t} = \left(E_{m\boldsymbol{k}} - E_{n\boldsymbol{k}'}\right)\eta^{(1)}_{m\boldsymbol{k};n\boldsymbol{k}'}(t) + \sum_{\mu\nu\boldsymbol{R}_1\boldsymbol{R}_2} \langle\psi_{m\boldsymbol{k}}|\mu\boldsymbol{R}_1\rangle Q^{(1)}_{\mu\boldsymbol{R}_1;\nu\boldsymbol{R}_2}(t) \langle\nu\boldsymbol{R}_2|\psi_{n\boldsymbol{k}'}\rangle.$$

Then, implementing the usual Fourier analysis via (19), we find

$$\eta^{(1)}_{m\boldsymbol{k};n\boldsymbol{k}'}(\omega) = -\sum_{\mu\nu\boldsymbol{R}_1\boldsymbol{R}_2} \frac{\langle\psi_{m\boldsymbol{k}}|\mu\boldsymbol{R}_1\rangle Q^{(1)}_{\mu\boldsymbol{R}_1;\nu\boldsymbol{R}_2}(\omega) \langle\nu\boldsymbol{R}_2|\psi_{n\boldsymbol{k}'}\rangle}{E_{m\boldsymbol{k}} - E_{n\boldsymbol{k}'} - \hbar(\omega + i0^+)},$$

where $0^+$ entering in the denominator describes the "turning on" of the electric field at $t > -\infty$. Finally, using the inverse of (A.6), we find

$$\eta^{(1)}_{\alpha\boldsymbol{R}'';\beta\boldsymbol{R}'}(\omega) = -\sum_{\mu\nu\boldsymbol{R}_1\boldsymbol{R}_2}\sum_{mn} \int_{\text{BZ}} d\boldsymbol{k}\, d\boldsymbol{k}'$$
$$\times \frac{\langle\alpha\boldsymbol{R}''|\psi_{m\boldsymbol{k}}\rangle \langle\psi_{m\boldsymbol{k}}|\mu\boldsymbol{R}_1\rangle Q^{(1)}_{\mu\boldsymbol{R}_1;\nu\boldsymbol{R}_2}(\omega) \langle\nu\boldsymbol{R}_2|\psi_{n\boldsymbol{k}'}\rangle \langle\psi_{n\boldsymbol{k}'}|\beta\boldsymbol{R}'\rangle}{E_{m\boldsymbol{k}} - E_{n\boldsymbol{k}'} - \hbar(\omega + i0^+)}. \tag{A.7}$$

Now, using the identity (10) and the result (A.5), we find

$$\sum_{\mu\nu\boldsymbol{R}_1\boldsymbol{R}_2} \int_{\text{BZ}} d\boldsymbol{k}\, d\boldsymbol{k}' \langle\alpha\boldsymbol{R}''|\psi_{m\boldsymbol{k}}\rangle \langle\psi_{m\boldsymbol{k}}|\mu\boldsymbol{R}_1\rangle Q^{(1)}_{\mu\boldsymbol{R}_1;\nu\boldsymbol{R}_2}(\omega) \langle\nu\boldsymbol{R}_2|\psi_{n\boldsymbol{k}'}\rangle \langle\psi_{n\boldsymbol{k}'}|\beta\boldsymbol{R}'\rangle$$
$$= -\frac{e\Omega_{uc}}{(2\pi)^d} E^l(\omega) \int_{\text{BZ}} d\boldsymbol{k}\, e^{i\boldsymbol{k}\cdot(\boldsymbol{R}''-\boldsymbol{R}')} U^{\dagger}_{\alpha m}(\boldsymbol{k})$$
$$\times \left[ f_{nm,\boldsymbol{k}}\left(\xi^l_{mn}(\boldsymbol{k}) + \mathcal{W}^l_{mn}(\boldsymbol{k})\right) + \left(i\delta_{nm}\partial_l f_{n\boldsymbol{k}} - f_{nm,\boldsymbol{k}}\mathcal{W}^l_{mn}(\boldsymbol{k})\right) \right] U_{n\beta}(\boldsymbol{k}),$$

where the first term in brackets results from the two terms in $Q^{(1)}_{\mu\boldsymbol{R}_1;\nu\boldsymbol{R}_2}(\omega)$ that involve dipole moments of the ELWFs, and the second term in brackets results from the term

$-eE^l(\omega)(R_1^l - R_2^l)\eta_{\mu R_1; \nu R_2}^{(0)}$ in $Q_{\mu R_1; \nu R_2}^{(1)}(\omega)$. In re-casting this second term as a single BZ integral, an integration by parts is performed and all surface terms are taken to vanish. While the integrand is periodic over BZ, it may not be smooth. Thus, this result is valid only if the ground state projector $\sum_n f_{n\mathbf{k}} |\psi_{n\mathbf{k}}\rangle \langle\psi_{n\mathbf{k}}|$ and therefore $\sum_n f_{n\mathbf{k}} U_{\alpha n}^\dagger(\mathbf{k}) U_{n\beta}(\mathbf{k})$ for any $\alpha$, $\beta$, which appears in the integrand, is smooth over BZ. While this is always true for insulators – topologically trivial or not – it is here an assumption. However, in the case of $p$-doped semiconductors considered here, we believe this to be valid if there are no degeneracies at the Fermi energy. With this we arrive at the result (20).

## B  Perturbation theory – An old-fashioned approach

Although Eq. (20) can be found as the extension of our earlier work presented in Appendix A, we believe some insight can be gained by looking at its derivation using a more traditional perturbation theory approach.

Consider first a molecule, where nuclei are considered fixed and the dynamics of the electron field operator $\hat{\psi}(\mathbf{x}, t)$ follows from the usual minimal coupling Hamiltonian,

$$\mathcal{H}_{\mathrm{mc}}(\mathbf{x}, t) = \frac{1}{2m}\left(\mathfrak{p}(\mathbf{x}) - \frac{e}{c}\mathbf{A}(\mathbf{x}, t)\right)^2 + V(\mathbf{x}) + e\phi(\mathbf{x}, t),$$

where $\mathfrak{p}(\mathbf{x})$ is given previously (7), the applied electromagnetic field is described by the scalar $\phi(\mathbf{x}, t)$ and vector $\mathbf{A}(\mathbf{x}, t)$ potentials, and $V(\mathbf{x})$ is the potential energy that confines the electrons to the nuclei. If the wavelength of light is much larger than the molecule, then the electric field $\mathbf{E}(\mathbf{x}, t)$ can be taken as uniform over the molecule, $\mathbf{E}(\mathbf{x}, t) = \mathbf{E}(t)$, and the magnetic field can be neglected. Via usual strategies [16], it can be shown that the dynamics of the electron field follows from the dipole Hamiltonian

$$\mathcal{H}_{\mathrm{dip}}(\mathbf{x}, t) = H_0(\mathbf{x}, \mathfrak{p}(\mathbf{x})) - e\mathbf{x} \cdot \mathbf{E}(t), \tag{B.1}$$

where $H_0(\mathbf{x}, \mathfrak{p}(\mathbf{x})) = \frac{1}{2m}(\mathfrak{p}(\mathbf{x}))^2 + V(\mathbf{x})$.

The use of (B.1) to describe instead the response of the electrons in an infinite crystal to long-wavelength radiation, where $\mathcal{H}_0(\mathbf{x})$ is now taken to be the Bloch Hamiltonian, is a strategy followed by Blount and others [33]; it has even been used to describe the nonlinear optical response of metals [37]. The appearance of a position operator in the interaction Hamiltonian requires calculations to be done cautiously, for giving meaning to matrix elements of the position with respect to the Bloch functions of the infinite crystal in a careful way is obviously problematic. In fact, the usual position operator is generally ill-defined to act on the Hilbert space containing such Bloch functions [9]. Moreover, it does not seem possible to implement a generalization of this kind of approach to treat instances where the electromagnetic field cannot be approximated as uniform. Indeed, that is one of the reasons the approach applied in this paper was developed. Nonetheless, this strategy does allow for the interaction Hamiltonian to be written as the sum of two terms, which can be identified as "interband" and "intraband." This permits the identification of the interband and intraband contributions to (20), at least within this perspective, and allows us to make contact with earlier work. And so we here present a derivation of (20) using this approach. Although most derivations [33] work in the electronic Hilbert space spanned by Bloch functions from the onset, some issues related to the position operator can be avoided if one works, at least initially, in an isomorphic Hilbert space spanned by a set of exponentially localized Wannier functions; the latter is a subspace of the space of square-integrable functions, where the usual position operator is well-defined. This is the approach we follow here. Moreover, we believe that this approach elucidates the physics of the two terms. Yet we ask the reader to forgive the mathematically questionable steps that

are part of the derivation and that are not characteristic of the rest of this paper. We feel that the cavalier approach we take in this Appendix is justified by the insight that the resulting expressions provide.

Working in the Heisenberg picture, the one-body operator on the electronic Fock space related to (B.1) is

$$\hat{H}(t) = \hat{H}_0(t) + \hat{V}_{\text{dip}}(t),$$ (B.2)

where

$$\hat{H}_0(t) \equiv \int \hat{\psi}^\dagger(\boldsymbol{x}, t) H_0(\boldsymbol{x}, \mathfrak{p}(\boldsymbol{x})) \hat{\psi}(\boldsymbol{x}, t) d\boldsymbol{x},$$

$$\hat{V}_{\text{dip}}(t) \equiv -eE^a(t) \int \hat{\psi}^\dagger(\boldsymbol{x}, t) x^a \hat{\psi}(\boldsymbol{x}, t) d\boldsymbol{x}.$$

The primary quantities of interest, the expectation values of the electronic charge and current density operators for a crystal initially occupying its $T = 0$ ground state $|\text{gs}\rangle$, can be extracted from the single-particle electron Green function

$$G(\boldsymbol{x}, \boldsymbol{y}; t) \equiv i \langle \text{gs}| \hat{\psi}^\dagger(\boldsymbol{y}, t) \hat{\psi}(\boldsymbol{x}, t) |\text{gs}\rangle.$$ (B.3)

We now move from the Heisenberg picture to the interaction picture, wherein operators on Fock space evolve under $\hat{H}_0$ and the effect of the perturbation is accounted for in the evolution of the electronic state $|\psi(t)\rangle = \hat{\mathcal{U}}(t)|\text{gs}\rangle$, where the time-evolution operator $\hat{\mathcal{U}}(t)$ is given by[17]

$$\hat{\mathcal{U}}(t) = 1 + \sum_{N=1}^{\infty} \int_{-\infty}^{t} \frac{dt_N}{i\hbar} \hat{V}_I(t_N) \cdots \int_{-\infty}^{t_2} \frac{dt_1}{i\hbar} \hat{V}_I(t_1),$$ (B.4)

for $\hat{V}_I(t) \equiv -eE^a(t) \int \hat{\psi}_0^\dagger(\boldsymbol{x}, t) x^a \hat{\psi}_0(\boldsymbol{x}, t) d\boldsymbol{x}$. The electron Green function (B.3) is then rewritten as

$$G(\boldsymbol{x}, \boldsymbol{y}; t) = i \langle \psi(t)| \hat{\psi}_0^\dagger(\boldsymbol{y}, t) \hat{\psi}_0(\boldsymbol{x}, t) |\psi(t)\rangle.$$ (B.5)

Noting that a (complete) set of ELWFs spans the single-particle electronic Hilbert space, the related operators can be used as a basis with respect to which the electron field operator $\hat{\psi}_0(\boldsymbol{x}, t)$ can be expanded,[18]

$$\hat{\psi}_0(\boldsymbol{x}, t) \equiv \sum_{\alpha \boldsymbol{R}} W_{\alpha \boldsymbol{R}}(\boldsymbol{x}) \hat{a}_{\alpha \boldsymbol{R}}(t),$$ (B.6)

where the operator $\hat{a}_{\alpha \boldsymbol{R}}^{(\dagger)}(t)$ here evolves as $i\hbar \frac{d}{dt} \hat{a}_{\alpha \boldsymbol{R}}^{(\dagger)}(t) = [\hat{a}_{\alpha \boldsymbol{R}}^{(\dagger)}(t), \hat{H}_0]$ and thus $\hat{a}_{\alpha \boldsymbol{R}}^{(\dagger)}(t) = e^{i\hat{H}_0 t/\hbar} \hat{a}_{\alpha \boldsymbol{R}}^{(\dagger)} e^{-i\hat{H}_0 t/\hbar}$. Then,

$$\hat{V}_I(t) = -e \frac{\Omega_{uc}}{(2\pi)^d} E^a(t) \sum_{\alpha\beta \boldsymbol{R}\boldsymbol{R}'} \left( \int_{\text{BZ}} d\boldsymbol{k} e^{i\boldsymbol{k}\cdot(\boldsymbol{R}-\boldsymbol{R}')} \tilde{\xi}_{\alpha\beta}^a(\boldsymbol{k}) \right) \hat{a}_{\alpha \boldsymbol{R}}^\dagger(t) \hat{a}_{\beta \boldsymbol{R}'}(t)$$

$$- eE^a(t) \sum_{\alpha \boldsymbol{R}} R^a \hat{a}_{\alpha \boldsymbol{R}}^\dagger(t) \hat{a}_{\alpha \boldsymbol{R}}(t).$$ (B.7)

---

[17]See, e.g., [46].
[18]See, e.g., [46].

Implementing (B.7) in (B.4) and using that result in (B.5), we find

$$G(\boldsymbol{x},\boldsymbol{y};t) = i \langle \text{gs}| \hat{\psi}_0^\dagger(\boldsymbol{y},t)\hat{\psi}_0(\boldsymbol{x},t)|\text{gs}\rangle + \frac{1}{\hbar} \int\limits_{-\infty}^{t} dt' \, \langle \text{gs}| \hat{\psi}_0^\dagger(\boldsymbol{y},t)\hat{\psi}_0(\boldsymbol{x},t)\hat{V}_I(t')|\text{gs}\rangle$$

$$- \frac{1}{\hbar} \int\limits_{-\infty}^{t} dt' \, \langle \text{gs}| \hat{V}_I^\dagger(t')\hat{\psi}_0^\dagger(\boldsymbol{y},t)\hat{\psi}_0(\boldsymbol{x},t)|\text{gs}\rangle + \dots$$

$$\equiv i \sum_{\mu \nu \boldsymbol{R}_1 \boldsymbol{R}_2} W_{\nu \boldsymbol{R}_2}(\boldsymbol{x})\Big(\eta^{(0)}_{\nu \boldsymbol{R}_1;\mu \boldsymbol{R}_2} + \eta^{(1)}_{\nu \boldsymbol{R}_1;\mu \boldsymbol{R}_2}(t) + \dots \Big) W^*_{\mu \boldsymbol{R}_1}(\boldsymbol{y}). \tag{B.8}$$

Note that in Eq. (36) of past work [16] we introduced the single-particle density matrix $\eta_{\alpha \boldsymbol{R};\beta \boldsymbol{R}'}$ such that it involved operators generating "adjusted Wannier functions" $\bar{W}_{\alpha \boldsymbol{R}}(\boldsymbol{x},t)$ (see Eq. (27, 30, 33) of Mahon *et al.* [16]) as well as a generalized Peierls phase $\Phi(\boldsymbol{x},\boldsymbol{y};t)$ (see Eq. (15) there). Thus, in general, it is not the minimal coupling Green function $G(\boldsymbol{x},\boldsymbol{y};t)$ to which $\eta_{\alpha \boldsymbol{R};\beta \boldsymbol{R}'}$ is "naturally" related, but rather the "global" Green function. However, in the case of a uniform electric field considered here, the corresponding vector potential $\boldsymbol{A}$ is necessarily uniform and $\Phi(\boldsymbol{x},\boldsymbol{y};t) = \frac{e}{\hbar c}(\boldsymbol{x}-\boldsymbol{y})\cdot \boldsymbol{A}(t)$ for a choice of straight-line path in the relators. Then, in this case, Eq. (32) of that work simplifies as

$$G(\boldsymbol{x},\boldsymbol{y};t) = i \sum_{\alpha \beta \boldsymbol{R} \boldsymbol{R}'} \bar{W}_{\alpha \boldsymbol{R}}(\boldsymbol{x},t)\breve{\eta}_{\alpha \boldsymbol{R};\beta \boldsymbol{R}'}(t)\bar{W}^*_{\beta \boldsymbol{R}'}(\boldsymbol{y},t)$$

$$= i e^{\boldsymbol{A}(t)\cdot(\boldsymbol{x}-\boldsymbol{y})} \sum_{\alpha \beta \boldsymbol{R} \boldsymbol{R}'} W_{\alpha \boldsymbol{R}}(\boldsymbol{x})\eta_{\alpha \boldsymbol{R};\beta \boldsymbol{R}'}(t)W^*_{\beta \boldsymbol{R}'}(\boldsymbol{y}). \tag{B.9}$$

In this Appendix we employ the gauge choice $\phi(\boldsymbol{x},t) = -\boldsymbol{x}\cdot \boldsymbol{E}(t)$, $\boldsymbol{A}(t) = \boldsymbol{0}$, such that the phase $\Phi(\boldsymbol{x},\boldsymbol{y};t)$ on the RHS of (B.9) vanishes. Thus, the identification of the single-particle density matrix in (B.8) is consistent with past work.

Now,

$$G_0(\boldsymbol{x},\boldsymbol{y};t) \equiv i \langle \text{gs}| \hat{\psi}_0^\dagger(\boldsymbol{y},t)\hat{\psi}_0(\boldsymbol{x},t)|\text{gs}\rangle = i \sum_{n} \int_{\text{BZ}} d\boldsymbol{k} f_{n\boldsymbol{k}} \psi^*_{n\boldsymbol{k}}(\boldsymbol{y})\psi_{n\boldsymbol{k}}(\boldsymbol{x})$$

$$= \sum_{\mu \nu \boldsymbol{R}_1 \boldsymbol{R}_2} W_{\nu \boldsymbol{R}_2}(\boldsymbol{x})\left(i\frac{\Omega_{uc}}{(2\pi)^d}\sum_{n}\int_{\text{BZ}} d\boldsymbol{k} f_{n\boldsymbol{k}} e^{i\boldsymbol{k}\cdot(\boldsymbol{R}_2-\boldsymbol{R}_1)}U^\dagger_{\nu n}(\boldsymbol{k})U_{n\mu}(\boldsymbol{k})\right) W^*_{\mu \boldsymbol{R}_1}(\boldsymbol{y}),$$

and we thus identify

$$\eta^{(0)}_{\nu \boldsymbol{R}_2;\mu \boldsymbol{R}_1} = \frac{\Omega_{uc}}{(2\pi)^d}\sum_{n}\int_{\text{BZ}} d\boldsymbol{k} f_{n\boldsymbol{k}} e^{i\boldsymbol{k}\cdot(\boldsymbol{R}_2-\boldsymbol{R}_1)}U^\dagger_{\nu n}(\boldsymbol{k})U_{n\mu}(\boldsymbol{k}), \tag{B.10}$$

yielding (18).

Next consider $\eta^{(1)}_{\nu R_2;\mu R_1}(t)$, which from (B.8) we identify as

$$
\eta^{(1)}_{\nu R_2;\mu R_1}(t) = \frac{ie}{\hbar} \frac{\Omega_{uc}}{(2\pi)^d} \int_{-\infty}^{t} dt' E^a(t') \sum_{\alpha\beta RR'} \int_{BZ} dk' e^{ik'\cdot(R-R')} \tilde{\xi}^a_{\alpha\beta}(k') \times
$$
$$
\left( \langle gs| \hat{a}^\dagger_{\mu R_1}(t)\hat{a}_{\nu R_2}(t)\hat{a}^\dagger_{\alpha R}(t')\hat{a}_{\beta R'}(t') |gs\rangle \right)
$$
$$
+ \frac{ie}{\hbar} \int_{-\infty}^{t} dt' E^a(t') \sum_{\alpha R} R^a \left( \langle gs| \hat{a}^\dagger_{\mu R_1}(t)\hat{a}_{\nu R_2}(t)\hat{a}^\dagger_{\alpha R}(t')\hat{a}_{\alpha R}(t') |gs\rangle \right)
$$
$$
- \frac{ie}{\hbar} \frac{\Omega_{uc}}{(2\pi)^d} \int_{-\infty}^{t} dt' E^a(t') \sum_{\alpha\beta RR'} \int_{BZ} dk' e^{ik'\cdot(R'-R)} \tilde{\xi}^a_{\beta\alpha}(k') \times
$$
$$
\left( \langle gs| \hat{a}^\dagger_{\beta R'}(t')\hat{a}_{\alpha R}(t')\hat{a}^\dagger_{\mu R_1}(t)\hat{a}_{\nu R_2}(t) |gs\rangle \right)
$$
$$
- \frac{ie}{\hbar} \int_{-\infty}^{t} dt' E^a(t') \sum_{\alpha R} R^a \left( \langle gs| \hat{a}^\dagger_{\alpha R}(t')\hat{a}_{\alpha R}(t')\hat{a}^\dagger_{\mu R_1}(t)\hat{a}_{\nu R_2}(t) |gs\rangle \right)
$$
$$
\equiv \eta^{(1;a)}_{\nu R_2;\mu R_1}(t) + \eta^{(1;b)}_{\nu R_2;\mu R_1}(t). \tag{B.11}
$$

We group the second and final lines of (B.11) into $\eta^{(1;a)}_{\nu R_2;\mu R_1}(t)$, and the first and third lines into $\eta^{(1;b)}_{\nu R_2;\mu R_1}(t)$. That is, $\eta^{(1;a)}_{\nu R_2;\mu R_1}(t)$ involves the contributions to $\eta^{(1)}_{\nu R_2;\mu R_1}(t)$ arising from the first term of (B.7), while $\eta^{(1;b)}_{\nu R_2;\mu R_1}(t)$ involves the contributions to $\eta^{(1)}_{\nu R_2;\mu R_1}(t)$ arising from the second term of (B.7). After some algebra we find

$$
\eta^{(1;a)}_{\nu R_2;\mu R_1}(t) \equiv \frac{ie}{\hbar} \int_{-\infty}^{t} dt' E^a(t') \sum_{\alpha R} R^a \times
$$
$$
\left( \langle gs| \hat{a}^\dagger_{\mu R_1}(t)\hat{a}_{\nu R_2}(t)\hat{a}^\dagger_{\alpha R}(t')\hat{a}_{\alpha R}(t') |gs\rangle - \langle gs| \hat{a}^\dagger_{\alpha R}(t')\hat{a}_{\alpha R}(t')\hat{a}^\dagger_{\mu R_1}(t)\hat{a}_{\nu R_2}(t) |gs\rangle \right)
$$
$$
= e \sum_{\omega} e^{-i(\omega+i0^+)t} E^a(\omega) \frac{\Omega_{uc}}{(2\pi)^d} \sum_{n} \int_{BZ} dk \times
$$
$$
\left( \frac{f_{nk} e^{ik\cdot(R_2-R_1)}}{\hbar(\omega+i0^+)} \left( (R_1^a - R_2^a) U^\dagger_{\nu n}(k)U_{n\mu}(k) + i\partial_a \left( U^\dagger_{\nu n}(k)U_{n\mu}(k) \right) \right) \right.
$$
$$
\left. - \sum_{m} \frac{f_{nm,k} e^{ik\cdot(R_2-R_1)}}{E_{mk} - E_{nk} - \hbar(\omega+i0^+)} U^\dagger_{\nu m}(k)\mathcal{W}^a_{mn}(k)U_{n\mu}(k) \right), \tag{B.12}
$$

where we have integrated by parts and taken any surface terms to vanish; this again demands smoothness of the integrand over BZ and requires the same assumption described in Appendix A. We have also taken the electric field $E(t)$ to be adiabatically applied at $t = -\infty$ resulting in the "$i0^+$" in the denominator and in the phase of $e^{-i(\omega+i0^+)t}$. We now consider $\eta^{(1;b)}_{\nu R_2;\mu R_1}(t)$. Using the completeness relation in the electronic Fock space

$$
1 = |gs\rangle \langle gs| + \sum_{cv} \int_{BZ} dk \, |cvk\rangle \langle cvk| + \dots, \tag{B.13}
$$

where $|cv\boldsymbol{k}\rangle \equiv \hat{a}_{c\boldsymbol{k}}^\dagger \hat{a}_{v\boldsymbol{k}} |\text{gs}\rangle$, $|cv\boldsymbol{k}, c_1 v_1 \boldsymbol{k}_1\rangle \equiv \hat{a}_{c\boldsymbol{k}}^\dagger \hat{a}_{v\boldsymbol{k}} \hat{a}_{c_1\boldsymbol{k}_1}^\dagger \hat{a}_{v_1\boldsymbol{k}_1} |\text{gs}\rangle$, etc., we find

$$
\eta_{v\boldsymbol{R}_2;\mu\boldsymbol{R}_1}^{(1;\text{b})}(t) \equiv \frac{ie}{\hbar} \frac{\Omega_{uc}}{(2\pi)^d} \int_{-\infty}^{t} dt' E^a(t') \sum_{\alpha\beta\boldsymbol{R}\boldsymbol{R}'} \int_{\text{BZ}} d\boldsymbol{k}' e^{i\boldsymbol{k}'\cdot(\boldsymbol{R}-\boldsymbol{R}')} \breve{\xi}_{\alpha\beta}^a(\boldsymbol{k}') \times
$$

$$
\left( \langle \text{gs}| \hat{a}_{\mu\boldsymbol{R}_1}^\dagger(t) \hat{a}_{v\boldsymbol{R}_2}(t) \hat{a}_{\alpha\boldsymbol{R}}^\dagger(t') \hat{a}_{\beta\boldsymbol{R}'}(t') |\text{gs}\rangle \right)
$$

$$
- \frac{ie}{\hbar} \frac{\Omega_{uc}}{(2\pi)^d} \int_{-\infty}^{t} dt' E^a(t') \sum_{\alpha\beta\boldsymbol{R}\boldsymbol{R}'} \int_{\text{BZ}} d\boldsymbol{k}' e^{i\boldsymbol{k}'\cdot(\boldsymbol{R}'-\boldsymbol{R})} \breve{\xi}_{\beta\alpha}^a(\boldsymbol{k}') \times
$$

$$
\left( \langle \text{gs}| \hat{a}_{\beta\boldsymbol{R}'}^\dagger(t') \hat{a}_{\alpha\boldsymbol{R}}(t') \hat{a}_{\mu\boldsymbol{R}_1}^\dagger(t) \hat{a}_{v\boldsymbol{R}_2}(t) |\text{gs}\rangle \right)
$$

$$
= e \frac{\Omega_{uc}}{(2\pi)^d} \sum_{\omega} e^{-i(\omega+i0^+)t} E^a(\omega) \int_{\text{BZ}} d\boldsymbol{k}' \times
$$

$$
\sum_{mn} f_{nm,\boldsymbol{k}'} \frac{e^{i\boldsymbol{k}'\cdot(\boldsymbol{R}_2-\boldsymbol{R}_1)} U_{vm}^\dagger(\boldsymbol{k}') \left( \xi_{mn}^a(\boldsymbol{k}') + \mathcal{W}_{mn}^a(\boldsymbol{k}') \right) U_{n\mu}(\boldsymbol{k}')}{E_{m\boldsymbol{k}'} - E_{n\boldsymbol{k}'} - \hbar(\omega+i0^+)} . \quad \text{(B.14)}
$$

Notably terms resulting from the first term of the completeness relation (B.13), which would involve diagonal matrix elements, cancel one another. Then combining (B.12) with (B.14) and implementing (19), we find

$$
\eta_{v\boldsymbol{R}_2;\mu\boldsymbol{R}_1}^{(1)}(\omega) = e \frac{\Omega_{uc}}{(2\pi)^d} E^a(\omega) \int_{\text{BZ}} d\boldsymbol{k} e^{i\boldsymbol{k}\cdot(\boldsymbol{R}_2-\boldsymbol{R}_1)} \sum_{mn} f_{nm,\boldsymbol{k}} \frac{U_{vm}^\dagger(\boldsymbol{k}) \xi_{mn}^a(\boldsymbol{k}) U_{n\mu}(\boldsymbol{k})}{E_{m\boldsymbol{k}} - E_{n\boldsymbol{k}} - \hbar(\omega+i0^+)}
$$

$$
+ e \frac{\Omega_{uc}}{(2\pi)^d} \frac{E^a(\omega)}{\hbar(\omega+i0^+)} \int_{\text{BZ}} d\boldsymbol{k} \sum_{n} e^{i\boldsymbol{k}\cdot(\boldsymbol{R}_2-\boldsymbol{R}_1)} f_{n\boldsymbol{k}} \times
$$

$$
\left( (R_1^a - R_2^a) U_{vn}^\dagger(\boldsymbol{k}) U_{n\mu}(\boldsymbol{k}) + i\partial_a \left( U_{vn}^\dagger(\boldsymbol{k}) U_{n\mu}(\boldsymbol{k}) \right) \right)
$$

$$
= e \frac{\Omega_{uc}}{(2\pi)^d} E^a(\omega) \int_{\text{BZ}} d\boldsymbol{k} e^{i\boldsymbol{k}\cdot(\boldsymbol{R}_2-\boldsymbol{R}_1)} \sum_{mn} f_{nm,\boldsymbol{k}} \frac{U_{vm}^\dagger(\boldsymbol{k}) \xi_{mn}^a(\boldsymbol{k}) U_{n\mu}(\boldsymbol{k})}{E_{m\boldsymbol{k}} - E_{n\boldsymbol{k}} - \hbar(\omega+i0^+)}
$$

$$
- ie \frac{\Omega_{uc}}{(2\pi)^d} \frac{E^a(\omega)}{\hbar(\omega+i0^+)} \int_{\text{BZ}} d\boldsymbol{k} \sum_{n} e^{i\boldsymbol{k}\cdot(\boldsymbol{R}_2-\boldsymbol{R}_1)} (\partial_a f_{n\boldsymbol{k}}) U_{vn}^\dagger(\boldsymbol{k}) U_{n\mu}(\boldsymbol{k}), \quad \text{(B.15)}
$$

where we have again used an integration by parts. Notably the term in (B.15) that diverges in the dc limit arises from the interaction term $-eE^a(t) \sum_{\alpha\boldsymbol{R}} R^a \hat{a}_{\alpha\boldsymbol{R}}^\dagger(t) \hat{a}_{\alpha\boldsymbol{R}}(t)$, the second term of (B.7). At first one might suspect that it is the sum over Bravais lattice vectors $\boldsymbol{R}$ that leads to the dc divergence, or if not, some other divergence. But, in fact, this is not the case because in the linear response calculation the relevant objects are of the form $\sum_{\boldsymbol{R}} R^a \langle \text{gs}| \hat{a}_{\mu\boldsymbol{R}_1}^\dagger(t) \hat{a}_{v\boldsymbol{R}_2}(t) \hat{a}_{\alpha\boldsymbol{R}}^\dagger(t') \hat{a}_{\alpha\boldsymbol{R}}(t') |\text{gs}\rangle$ (see the first equality of (B.12)); thus not all $\boldsymbol{R}$'s contribute equally and the result of such a sum appears to be finite.

To gain further insight into origin of the terms appearing in (B.15), it is useful to rewrite $\hat{V}_{\text{I}}(t)$ in terms of the operators that generate the single-particle Bloch energy eigenvectors. The second term of (B.7) involves

$$
\sum_{\alpha\boldsymbol{R}} R^a \hat{a}_{\alpha\boldsymbol{R}}^\dagger(t) \hat{a}_{\alpha\boldsymbol{R}}(t) = \frac{i}{2} \int_{\text{BZ}} d\boldsymbol{k} \sum_{n} \left( \hat{a}_{n\boldsymbol{k}}^\dagger(t) \left( \partial_a \hat{a}_{n\boldsymbol{k}}(t) \right) - \left( \partial_a \hat{a}_{n\boldsymbol{k}}^\dagger(t) \right) \hat{a}_{n\boldsymbol{k}}(t) \right)
$$

$$
- \int_{\text{BZ}} d\boldsymbol{k} \sum_{nm} \hat{a}_{n\boldsymbol{k}}^\dagger(t) \mathcal{W}_{nm}(\boldsymbol{k}) \hat{a}_{m\boldsymbol{k}}(t). \quad \text{(B.16)}
$$

When implemented in the linear response calculation, the first two terms of (B.16) give non-zero contributions only for those $\boldsymbol{k}$ "near" the Fermi surface and indeed gives vanishing contribution if $|\text{gs}\rangle$ is the ground state of a trivial insulator. That such an interaction term leads

to a diverging induced free current density is in-line with physical expectation. The first term of (B.7) can also be rewritten,

$$
\sum_{\alpha\beta RR'} \hat{a}^\dagger_{\alpha R}(t)\hat{a}_{\beta R'}(t) \int_{\mathrm{BZ}} d\boldsymbol{k}\, e^{i\boldsymbol{k}\cdot(\boldsymbol{R}-\boldsymbol{R'})}\tilde{\xi}^a_{\alpha\beta}(\boldsymbol{k})
$$
$$
= \frac{(2\pi)^d}{\Omega_{uc}} \int_{\mathrm{BZ}} d\boldsymbol{k} \sum_{nm} \hat{a}^\dagger_{n\boldsymbol{k}}(t)\Big(\xi^a_{nm}(\boldsymbol{k}) + \mathcal{W}^a_{nm}(\boldsymbol{k})\Big)\hat{a}_{m\boldsymbol{k}}(t). \tag{B.17}
$$

The net result is

$$
\hat{V}_I(t) = -e\frac{\Omega_{uc}}{(2\pi)^d}E^a(t) \sum_{\alpha\beta RR'} \hat{a}^\dagger_{\alpha R}(t)\hat{a}_{\beta R'}(t) \int_{\mathrm{BZ}} d\boldsymbol{k}\, e^{i\boldsymbol{k}\cdot(\boldsymbol{R}-\boldsymbol{R'})}\tilde{\xi}^a_{\alpha\beta}(\boldsymbol{k})
$$
$$
- eE^a(t)\sum_{\alpha R} R^a \hat{a}^\dagger_{\alpha R}(t)\hat{a}_{\alpha R}(t)
$$
$$
= -eE^a(t) \int_{\mathrm{BZ}} d\boldsymbol{k} \sum_{nm} \hat{a}^\dagger_{n\boldsymbol{k}}(t)\xi^a_{nm}(\boldsymbol{k})\hat{a}_{m\boldsymbol{k}}(t)
$$
$$
+ \frac{ie}{2}E^a(t) \int_{\mathrm{BZ}} d\boldsymbol{k} \sum_{n} \Big(\big(\partial_a \hat{a}^\dagger_{n\boldsymbol{k}}(t)\big)\hat{a}_{n\boldsymbol{k}}(t) - \hat{a}^\dagger_{n\boldsymbol{k}}(t)\big(\partial_a \hat{a}_{n\boldsymbol{k}}(t)\big)\Big),
$$

which is gauge independent, as expected. As described above, due to the relative negative sign between terms involving $\hat{V}_I(t)$ and $\hat{V}^\dagger_I(t)$ in the perturbative expansion of the electron Green function, the interaction term involving $E^a(t)\int_{\mathrm{BZ}} d\boldsymbol{k}\sum_{nm}\hat{a}^\dagger_{n\boldsymbol{k}}(t)\xi^a_{nm}(\boldsymbol{k})\hat{a}_{m\boldsymbol{k}}(t)$ gives rise only to terms for which $n \neq m$, which we refer to as being related to the "interband response" (see, for example, the cancellation of "intraband" terms in (B.14)). In contrast, we refer to the terms resulting from the interaction term involving $iE^a(t)\int_{\mathrm{BZ}} d\boldsymbol{k}\sum_{n}\Big(\hat{a}^\dagger_{n\boldsymbol{k}}(t)\big(\partial_a\hat{a}_{n\boldsymbol{k}}(t)\big) - \big(\partial_a\hat{a}^\dagger_{n\boldsymbol{k}}(t)\big)\hat{a}_{n\boldsymbol{k}}(t)\Big)$ as being related to the "intraband response."

## C Time-reversal symmetry

Taking $\mathcal{T}\,|\psi_{n\boldsymbol{k}}\rangle \overset{\mathcal{T}}{=} e^{-i\lambda_n(\boldsymbol{k})}\,|\psi_{n-\boldsymbol{k}}\rangle$ [2], which is equivalent to $\psi^*_{n\boldsymbol{k}}(\boldsymbol{x}) = \mathcal{T}\psi_{n\boldsymbol{k}}(\boldsymbol{x}) \overset{\mathcal{T}}{=} e^{-i\lambda_n(\boldsymbol{k})}\psi_{n-\boldsymbol{k}}(\boldsymbol{x})$, or alternatively $u^*_{n\boldsymbol{k}}(\boldsymbol{x}) \overset{\mathcal{T}}{=} e^{-i\lambda_n(\boldsymbol{k})}u_{n-\boldsymbol{k}}(\boldsymbol{x})$, yields

$$
\xi^a_{nm}(\boldsymbol{k}) \overset{\mathcal{T}}{=} e^{i(\lambda_m(\boldsymbol{k})-\lambda_n(\boldsymbol{k}))}\xi^a_{mn}(-\boldsymbol{k}) - \delta_{nm}\frac{\partial\lambda_m(\boldsymbol{k})}{\partial k^a}, \tag{C.1}
$$

and as well $E_{n\boldsymbol{k}} \overset{\mathcal{T}}{=} E_{n-\boldsymbol{k}}$, which implies $f_{n\boldsymbol{k}} \overset{\mathcal{T}}{=} f_{n-\boldsymbol{k}}$. Furthermore, time-reversal symmetry allows the ELWFs to be chosen such that they are real-valued functions [19, 38], and taking $W_{\alpha R}(\boldsymbol{x}) \overset{\mathcal{T}}{=} W^*_{\alpha R}(\boldsymbol{x})$ yields

$$
U_{n\alpha}(\boldsymbol{k}) \overset{\mathcal{T}}{=} U^\dagger_{\alpha n}(-\boldsymbol{k})e^{-i\lambda_n(-\boldsymbol{k})},
$$

which leads to

$$
\mathcal{W}^a_{nm}(\boldsymbol{k}) \overset{\mathcal{T}}{=} e^{i(\lambda_m(-\boldsymbol{k})-\lambda_n(-\boldsymbol{k}))}\mathcal{W}^a_{mn}(-\boldsymbol{k}) - \delta_{nm}\frac{\partial\lambda_n(-\boldsymbol{k})}{\partial(-k)^i}. \tag{C.2}
$$

With these relations one can show

$$\int_{\text{BZ}} \frac{d\mathbf{k}}{(2\pi)^d} \sum_n f_{n\mathbf{k}} \partial_l \left( \xi_{nn}^i(\mathbf{k}) + \mathcal{W}_{nn}^i(\mathbf{k}) \right) \overset{\mathcal{T}}{=} -\int_{\text{BZ}} \frac{d\mathbf{k}}{(2\pi)^d} \sum_n f_{n\mathbf{k}} \partial_l \left( \xi_{nn}^i(\mathbf{k}) + \mathcal{W}_{nn}^i(\mathbf{k}) \right),$$

and therefore vanishes. It then immediately follows that $J_B^{(1)}(\omega = 0) \overset{\mathcal{T}}{=} \mathbf{0}$, or equivalently that the term in (28) that diverges in the dc limit vanishes. Moreover from the relations (C.1,C.2) it follows that $\mathbf{M}^{(0)} \overset{\mathcal{T}}{=} \mathbf{0}$.

## D  Link currents and the related free current density

Recall from past work [16] that in the "long-wavelength limit"

$$H_{\alpha\mathbf{R}'';\lambda\mathbf{R}'}(\omega) = \int W_{\alpha\mathbf{R}''}^*(\mathbf{x}) H_0(\mathbf{x}, \mathfrak{p}(\mathbf{x})) W_{\lambda\mathbf{R}'}(\mathbf{x}) d\mathbf{x}$$

$$- \frac{e}{2} \int W_{\alpha\mathbf{R}''}^*(\mathbf{x}) \left( (\mathbf{x} - \mathbf{R}'') + (\mathbf{x} - \mathbf{R}') \right) \cdot \mathbf{E}(t) W_{\lambda\mathbf{R}'}(\mathbf{x}) d\mathbf{x}, \tag{D.1}$$

and since we write $H_{\alpha\mathbf{R}'';\lambda\mathbf{R}'}(\omega) = H_{\alpha\mathbf{R}'';\lambda\mathbf{R}'}^{(0)} + H_{\alpha\mathbf{R}'';\lambda\mathbf{R}'}^{(1)}(\omega)$, with all higher order contributions vanishing in this case, we identify

$$H_{\alpha\mathbf{R}'';\lambda\mathbf{R}'}^{(0)} = \int W_{\alpha\mathbf{R}''}^*(\mathbf{x}) H_0(\mathbf{x}, \mathfrak{p}(\mathbf{x})) W_{\lambda\mathbf{R}'}(\mathbf{x}) d\mathbf{x}, \tag{D.2}$$

$$H_{\alpha\mathbf{R}'';\lambda\mathbf{R}'}^{(1)}(\omega) = -\frac{e}{2} \int W_{\alpha\mathbf{R}''}^*(\mathbf{x}) \left( (\mathbf{x} - \mathbf{R}'') + (\mathbf{x} - \mathbf{R}') \right) \cdot \mathbf{E}(t) W_{\lambda\mathbf{R}'}(\mathbf{x}) d\mathbf{x}. \tag{D.3}$$

With this we implement the definition of $I(\mathbf{R}, \mathbf{R}'; \omega)$ previously given, and with (20) we find

$$I^{(1)}(\mathbf{R}, \mathbf{R}'; \omega)$$
$$= \frac{e}{i\hbar} \sum_{\alpha\lambda} \left( H_{\alpha\mathbf{R};\lambda\mathbf{R}'}^{(1)}(\omega) \eta_{\lambda\mathbf{R}';\alpha\mathbf{R}}^{(0)} - \eta_{\alpha\mathbf{R};\lambda\mathbf{R}'}^{(0)} H_{\lambda\mathbf{R}';\alpha\mathbf{R}}^{(1)}(\omega) \right)$$
$$+ \frac{e}{i\hbar} \sum_{\alpha\lambda} \left( H_{\alpha\mathbf{R};\lambda\mathbf{R}'}^{(0)} \eta_{\lambda\mathbf{R}';\alpha\mathbf{R}}^{(1)}(\omega) - \eta_{\alpha\mathbf{R};\lambda\mathbf{R}'}^{(1)}(\omega) H_{\lambda\mathbf{R}';\alpha\mathbf{R}}^{(0)} \right)$$
$$= -\frac{2e^2}{\hbar} \left( \frac{\Omega_{uc}}{(2\pi)^d} \right)^2 E^l(\omega) \sum_{\alpha\lambda} \int_{\text{BZ}} d\mathbf{k} d\mathbf{k}' \text{Im} \left[ e^{i(\mathbf{k}-\mathbf{k}')\cdot(\mathbf{R}-\mathbf{R}')} \sum_n f_{n\mathbf{k}'} U_{n\alpha}(\mathbf{k}') \tilde{\xi}_{\alpha\lambda}^l(\mathbf{k}) U_{\lambda n}^\dagger(\mathbf{k}') \right]$$
$$+ \frac{e^2}{i\hbar} \left( \frac{\Omega_{uc}}{(2\pi)^d} \right)^2 E^l(\omega) \sum_{\alpha\lambda} \int_{\text{BZ}} d\mathbf{k} \sum_s \int_{\text{BZ}} d\mathbf{k}' E_{s\mathbf{k}}$$
$$\times \left( e^{i(\mathbf{k}-\mathbf{k}')\cdot(\mathbf{R}-\mathbf{R}')} U_{s\lambda}(\mathbf{k}) \sum_{mn} \frac{f_{nm,\mathbf{k}'} U_{\lambda m}^\dagger(\mathbf{k}') \xi_{mn}^l(\mathbf{k}') U_{n\alpha}(\mathbf{k}')}{E_{m\mathbf{k}'} - E_{n\mathbf{k}'} - \hbar(\omega + i0^+)} U_{\alpha s}^\dagger(\mathbf{k}) \right.$$
$$\left. - e^{-i(\mathbf{k}-\mathbf{k}')\cdot(\mathbf{R}-\mathbf{R}')} U_{s\alpha}(\mathbf{k}) \sum_{mn} \frac{f_{nm,\mathbf{k}'} U_{\alpha m}^\dagger(\mathbf{k}') \xi_{mn}^l(\mathbf{k}') U_{n\lambda}(\mathbf{k}')}{E_{m\mathbf{k}'} - E_{n\mathbf{k}'} - \hbar(\omega + i0^+)} U_{\lambda s}^\dagger(\mathbf{k}) \right)$$
$$+ \frac{e^2}{i\hbar} \left( \frac{\Omega_{uc}}{(2\pi)^d} \right)^2 E^l(\omega) \sum_{\alpha\lambda} \int_{\text{BZ}} d\mathbf{k} \sum_s \int_{\text{BZ}} d\mathbf{k}' \frac{E_{s\mathbf{k}}}{\hbar(\omega + i0^+)}$$
$$\times \left( e^{i(\mathbf{k}-\mathbf{k}')\cdot(\mathbf{R}-\mathbf{R}')} U_{s\lambda}(\mathbf{k}) \sum_n f_{n\mathbf{k}'} \left( (R^l - R'^l) U_{\lambda n}^\dagger(\mathbf{k}') U_{n\alpha}(\mathbf{k}') + i\partial_l \left( U_{\lambda n}^\dagger(\mathbf{k}') U_{n\alpha}(\mathbf{k}') \right) \right) U_{\alpha s}^\dagger(\mathbf{k}) \right.$$
$$\left. - e^{-i(\mathbf{k}-\mathbf{k}')\cdot(\mathbf{R}-\mathbf{R}')} U_{s\alpha}(\mathbf{k}) \sum_n f_{n\mathbf{k}'} \left( (R'^l - R^l) U_{\alpha n}^\dagger(\mathbf{k}') U_{n\lambda}(\mathbf{k}') + i\partial_l \left( U_{\alpha n}^\dagger(\mathbf{k}') U_{n\lambda}(\mathbf{k}') \right) \right) U_{\lambda s}^\dagger(\mathbf{k}) \right). \tag{D.4}$$

The first line of both equalities of (D.4) is the result of a "compositional" modification, while the remainder is the result of a "dynamical" modification; the second term of (31) is the result of the first term of (20) and the final term of (31) is the result of the second term of (20). Notably the first line of (31) is independent of energy and involves frequency only through $E(\omega)$, while this is generally not the case for the other terms.

In Sec. 4 we are interested, among other things, in the macroscopic free current density, $J_F(x,\omega)$, related to the microscopic free current density $j_F(x,\omega)$. In past work [17] we have described this averaging procedure in some detail, in particular for the microscopic polarization and magnetization fields. In the limit of a uniform applied electric field, the expressions Eq. (7), (9), (B4)-(B6), and (B8) presented there result in the macroscopic polarization and magnetization fields being uniform, and the only contributions being the dipole moments, (21). We here focus on the macroscopic free current density found by implement a spatial averaging function $w(x)$ to relate the microscopic and macroscopic quantities. That is,

$$J_F(x,\omega) \equiv \int w(x-x')j_F(x',\omega)dx'. \tag{D.5}$$

Implementing the definition (30), the relator expansion [17]

$$s^i(w;x,y) \simeq (x^i-y^i)\delta(w-y) - \frac{1}{2}(x^i-y^i)(x^j-y^j)\frac{\partial\delta(w-y)}{\partial w^j} + \dots, \tag{D.6}$$

and noting that the first-order modification to the link currents here takes the form $I^{(1)}(R,R';\omega) = I^{(1)}(R-R',\omega)$, we find

$$
\begin{aligned}
J_F^{i(1)}(x,\omega) &= \frac{1}{2}\sum_{RR'}I^{(1)}(R-R',\omega) \\
&\quad \times \left((R^i-R'^i)w(x-R) + \frac{1}{2}(R^i-R'^i)(R^j-R'^j)\frac{\partial w(x-R)}{\partial x^j} + \dots\right) \\
&= \frac{1}{2}\sum_{R_1}I^{(1)}(R_1,\omega)\left(R_1^i\sum_R w(x-R) + \frac{1}{2}R_1^i R_1^j\frac{\partial}{\partial x^j}\sum_R w(x-R) + \dots\right) \\
&= \frac{1}{2\Omega_{uc}}\sum_{R_1}I^{(1)}(R_1,\omega)R_1^i, \tag{D.7}
\end{aligned}
$$

where in going to the final line we have used the special case of a uniform applied electric field in Eq. (B8) of [17]. Thus, we arrive at (32).

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
