# Peer review of "Electric polarization and magnetization in metals"

_SciPost Physics, doi:SciPost Phys. 14, 058 (2023)_

## Round 1 · Referee Report · Anonymous (Referee 1) · 2022-8-4

Strengths
Weaknesses
Report
Dear editor,
The authors propose a new definition of the electric polarization and magnetization. The authors claim that, contrary to previous definitions of these quantities (in particular, in the modern theories of polarization and magnetization), the new definitions are also well-defined for metals. My main criticism of this work is the absence of any calculation (analytical or numerical) that would support this claim and also, more generally, that would confirm the viability of the authors’ approach for practical calculations. I note that also in other recent works by the same authors (PRB 99 235140, PRR 2 033126, PRR 2 043110) in which they present the same general method I have not found any calculation. Would it be possible to add some simple calculation, e.g. tight-binding or other, to illustrate the authors’ approach? As is well-known several complications might arise when putting a new method to the test.
Please find some further questions and comments below
1) The main results of the paper seem to be the definition of the polarization in Eq. (26) and the definition of the magnetization in Eq. (27). Is this indeed the case?
2) The only thing that can be measured in experiment is the change of polarization which is fully defined in terms of the time integral of the current density (assuming there is no magnetization). This works equally well for insulators and for metals. So, what is the importance of defining an absolute polarization which is a gauge-dependent quantity? (see also point 7 below)
3) In the abstract the authors write “If instead one takes the view that the electronic polarization is more fundamentally related to the existence of a complete set of exponentially localized Wannier functions, a definition is always admitted.” I do not see what is fundamental about the existence of a complete set of exponentially localized Wannier functions (ELWF). They might be very useful in practice for numerical calculations and/or for interpretation of the results but what is fundamental about them?
4) The authors use site-based quantities such as a site density and site current density but they are never explicitly defined. Could the authors please add these definitions (or point to the appropriate equations in earlier works)? Are the definitions general or are they contingent on the choice of using ELWF?
5) In the introduction it is written that a lattice sums are performed to give the microscopic polarization and magnetization. Lattice sums are generally conditionally convergent which could lead to convergence problems in practical calculations. Could the authors comment on this?
6) Also in the introduction, the authors write “One reason is that usual calculations made in minimal coupling can require the identification of sum rules to show properly behaved results at low frequencies - especially if nonlinear optical response is calculated, which is a future direction for this work - and that is not a difficulty with the calculations presented here, since the response is calculated as due to electric and magnetic fields.” In the linear response the problem of divergencies at low frequencies have been addressed in the literature and several solutions have been proposed (PRB 82 035104, PRB 95 155203). Indeed, I think the use of sum rules is an elegant approach to avoid divergencies.
7) Just below the authors write “A second reason is that in an insulator there is a clear physical significance to the response of the polarization to applied fields, as has been demonstrated within the “modern theory,"” Could the authors explain what this physical significance is?
8) Why do the authors neglect local-field corrections which can be very important? Moreover, no calculations are performed in this work, everything is purely formal, so why neglect them?
9) The authors write “Consideration of the phenomena related to spatially-varying electromagnetic fields is left for future work.” but one page further they write “The primary difference between our approach and those is that this investigation is a limiting case of a more general framework within which spatial and temporal variation of electric and magnetic fields can be into account; that is not the case in earlier works.” If this is the primary difference it would seem important to not defer spatially-varying electromagnetic fields for future work but to include them here.
10) In the conclusions the authors write “A positive feature of the approach implemented here is that it captures the complete finite-frequency response of a metallic crystal; there is no need to implement distinct approaches to calculate the anomalous Hall and Drude contributions to J(1).” Could the authors compare to PRB 86 125139 which also seems to include all contributions. Drude contributions are also included in PRB 74 245117 but not anomalous Hall contributions because time-reversal symmetry is assumed. However, I would expect that also this contribution is present if this assumption is removed.
11) In appendix B the authors write “Although most derivations [43] work with Bloch functions from the onset, some issues related to the position operator can be avoided if one works, at least initially, in the Hilbert space containing ELWFs, the space of square-integrable functions, where the usual position operator is well-defined.” The usual position operator is not compatible with periodic boundary conditions (PBC) and, therefore, ill-defined whenever PBC are applied. Therefore, I do not understand why the authors claim that by invoking ELWFs the usual position operator is well-defined. Could the authors explain this point in more detail? I note that recently a position operator has been defined that is compatible with PBC (PRB 99 205144, PRB 105 235201).
Some minor points
12) The authors write “However, often times, and as we will take to be the case in this paper, a number of Hilbert subbundles of the Bloch bundle are trivializable, …”. What is meant by “trivializable” ?
13) The conclusion section is almost two pages long. The authors might consider to be more concise.
14) References [25] and [60] are the same.

Claudio Attaccalite on 2022-09-05 [id 2789]
Dear authors
while I'm searching for an additional referee, I would like to ask you some questions on your manuscript, that you can answer at the same time as your answer to the referees
1) the use of Wannier Functions makes difficult the application to the electron dynamics, because it is difficult to generate ELWF for de-localized conduction bands. May the authors comment on this point. Moreover, is it possible to reformulate the present result with non-orthogonal localized basis set?
2) In Ref. 5 a generalization of Magnetization for metals and topological insulators is proposed, do the authors compared their results with that formula?
3) Is it not possible to recast all equations in the Bloch space? At the end there is an equivalence between Wannier and Bloch space. In this case it will much easy to see the difference with standard literature.
4) In the introduction the authors, say that it is difficult to take into account "variation of the optical fields over a unit cell". I do not see many difficulties to treat fields that are not uniform, in the Bloch formalism, it is sufficient that the momentum of the field to be compatible q=k-k', similar to what is done with phonons calculations. Even if I agree that a formalism in ELWFs is more general and could thread both cases q=0 and q/=0.
5) Does this formalism go again the Kohn argument of localization in an insulator? Is possible to extend the present ideas to a correlated system?
6) Is it possible to quantify in model or a simple calculation the differences with Modern Theory of Polarization in the case a Chern insulator ? is there a way to compare this formalism with experimental measurements?

---

## Round 2 · Referee Report · Anonymous (Referee 2) · 2022-12-9

Report

Dear editor,

I thanks the authors for their answers to my questions and comments and for the revised manuscript. I think the manuscript can be accepted as an article in SciPost Physics if the authors take into account the following questions and comments

1)

In their answer the authors write

“We do agree with the referee that model calculations implementing our formalism would provide some physical insight. However, because ELWFs are central to our formalism, and because such functions are not provided in simple tight-binding models appearing in the literature (except, e.g., approximating them as delta functions), there is no simple calculation we can do at this time. “

However in the Introduction they write

“And a fifth reason is that, with its emphasis on ELWFs and the interest in those functions for electronic structure and response calculations in general, we can hope that the approach here will be useful in numerical calculations.”

These the remarks seem to be in contradiction.

Either ELWF are simple to implement and then the authors could have done so to add some numerical results to this work or they are not simple to implement and the usefulness of the authors’ approach is in doubt.
It would be good if the authors clearly state in the manuscript how their approach can be used in practice and what would be the difficulties for a practical implementation in a computer code.

2)

In the Introduction the authors write

“Here, polarization and magnetization fields, and free charge and current densities serve as intermediary quantities that aid calculation and provide physical insight, but in general only the appropriate combinations that lead to the charge and current densities have direct physical significance.”

Could the authors please write explicitly what this physical insight is ?

3)

The authors write

"One reason is that usual calculations made in minimal coupling can require the identification of sum rules to show properly behaved results at low frequencies [23, 24]"

I think Refs 23 and 24 should (at least) be complemented with PRB 82 035104 and PRB 95 155203

4) In their answer to my question 7 the authors write

“The physical significance here is that, in an ordinary insulator, the entire electronic response to an electric field is due to a modification of the electric polarization about each lattice site, and this is gauge invariant."

The electric polarisation around a lattice site is not a measurable quantity.
So what does “gauge invariance” mean ?

5) In their answer to my question 8 the authors write

“We agree that such corrections can be important. In this first work, we neglect such corrections to give room to fully analyze this more simple case, which is indeed not so very simple.”

I think the authors should mention this in the manuscript.

6) In their answer to my question 10 the authors write

"We do not claim this feature is unique to our method, only that it is a supporting aspect of it. We emphasize that consistency with works such as those referenced by the referee add support to the definitions we employ, as we arrive at the expected result via a new method. We have added text to the Introduction and refined the Conclusion to make this point more clear. For example, at the end of the Introduction we add the text beginning with “Ultimately, when implemented in a metallic crystal, that our general definitions agree with past work…”.”

I think the authors should (at least) mention PRB 86 125139

7) In their answer to my question 11 the authors write

“In a subspace of the Hilbert space of square-integrable functions, the position operator is well-defined (e.g., the usual position expectation value in a space of atomic wavefunctions is well-defined). A basis of ELWFs spans such a subspace, while the Bloch functions are not even square-integrable. This allows the position matrix elements in the basis of ELWFs to be well-defined, in particular since they have compact support; See, e.g., Ref [43].”

Do I then correctly understand that the authors’ approach does not use PBC?
How then do they propose to model a solid?
Could the authors clarify?

  • validity: -
  • significance: -
  • originality: -
  • clarity: -
  • formatting: -
  • grammar: -

Author:  Perry Mahon  on 2022-12-16  [id 3142]

(in reply to Report 1 on 2022-12-09)
Category:
answer to question

Please see the attached pdf for a reply to the referees comments.

Attachment:

Metals_SciPost_2.pdf

---

## Round 2 · Author Response

Dear Dr. Attaccalite,

We kindly thank both the referee and yourself for the many insightful comments. Please find below a series of responses to the questions posed. Also, given your suggestions we have significantly reworked the Abstract, Introduction, and Conclusion of the manuscript in the hope that our primary message is now more clear, and that any confusion regarding our approach is lifted. We hope that with these changes our manuscript can be published.

Best, PTM and JES

Reply to the editor:

Dear authors

while I'm searching for an additional referee, I would like to ask you some questions on your manuscript, that you can answer at the same time as your answer to the referees

1) the use of Wannier Functions makes difficult the application to the electron dynamics, because it is difficult to generate ELWF for de-localized conduction bands. May the authors comment on this point. Moreover, is it possible to reformulate the present result with non-orthogonal localized basis set?

As far as we are aware, it is generally possible to construct ELWFs for any set of isolated energy bands given that the corresponding bundle structure is trivial (see, e.g., Ref [30]); each Bloch energy eigenfunction is typically just as delocalized as the next. It is possible to use a non-orthogonal basis set, but we have not yet worked out the form of the expressions in such a basis.

2) In Ref. 5 a generalization of Magnetization for metals and topological insulators is proposed, do the authors compared their results with that formula?

We do compare our results with that generalization for the metallic case. We have added some words to the Abstract and Introduction and refined the Conclusion to make this comparison more clear. In the Abstract we now write, “However, the modern theory of magnetization has been extended via thermodynamic arguments to include metals and Chern insulators. We here compare with that generalization and find disagreement; the manner in which the expressions differ elucidates the distinct philosophies of these approaches.” In the Introduction we have added a paragraph beginning with “A set {…} that satisfies (1) …”. And in the Conclusion we have refined the paragraph beginning with “Although the expressions we obtain for…”.

3) Is it not possible to recast all equations in the Bloch space? At the end there is an equivalence between Wannier and Bloch space. In this case it will much easy to see the difference with standard literature.

We believe the form you seek are the BZ integrals Eq. (26,29,30,34,35). Indeed, in the text surrounding these equations (and elsewhere) we discuss comparison with the existing literature. This issue is addressed in the new text explicitly mentioned in response to your comment 2).

4) In the introduction the authors, say that it is difficult to take into account "variation of the optical fields over a unit cell". I do not see many difficulties to treat fields that are not uniform, in the Bloch formalism, it is sufficient that the momentum of the field to be compatible q=k-k', similar to what is done with phonons calculations. Even if I agree that a formalism in ELWFs is more general and could thread both cases q=0 and q/=0.

We agree, there is more than one method to perform such calculations. We suggest that our method can provide more physical insight than those other methods. Indeed, in past work we make contact with such calculations and add the sentence “Moreover, in said calculation we find agreement with the usual approach involving a q-expansion of the conductivity tensor [20].” to the Introduction.

5) Does this formalism go again the Kohn argument of localization in an insulator? Is possible to extend the present ideas to a correlated system?

This formalism follows a different line of reasoning, whereby we do not focus on localization properties of the ground state, but rather rely on the general existence of a complete set of ELWFs (subsets of which might be completely “occupied” or completely “unoccupied” in the insulating case, or be partially occupied, as is the case here) with respect to which our definitions are formulated. Yes, we believe that these ideas can be extended to interacting systems. This would be more straightforward in Hartree-Fock approximation, but also possible more generally. We intend to focus on such considerations in future work.

6) Is it possible to quantify in model or a simple calculation the differences with Modern Theory of Polarization in the case a Chern insulator ? is there a way to compare this formalism with experimental measurements?

The investigation of a Chern insulator is presented in a related work (available here: https://arxiv.org/abs/2203.05077), wherein we find an expression for the polarization similar to that of the modern theory. Regarding comparison to experiment, the physically accessible quantities need be gauge invariant, and here happen to be only the electronic charge and current densities. Then, at this level of calculation, the experimental measurements would be the usual quantum anomalous Hall and Drude conductivities. While these are of course well-known phenomena, that we can derive the expected form of the conductivity tensor as a sum of induced polarization and free current density provides support for our method. We add text in the Abstract and Introduction, and refine the Conclusion to make this point more clear. In the Abstract we modify the text beginning with “Our approach leads to the usual electrical conductivity tensor in the long-wavelength limit…”. At the end of the Introduction we add the text beginning with “Ultimately, when implemented in a metallic crystal, that our general definitions agree …”.

Reply to the referee:

Dear editor,

The authors propose a new definition of the electric polarization and magnetization. The authors claim that, contrary to previous definitions of these quantities (in particular, in the modern theories of polarization and magnetization), the new definitions are also well-defined for metals. My main criticism of this work is the absence of any calculation (analytical or numerical) that would support this claim and also, more generally, that would confirm the viability of the authors’ approach for practical calculations. I note that also in other recent works by the same authors (PRB 99 235140, PRR 2 033126, PRR 2 043110) in which they present the same general method I have not found any calculation. Would it be possible to add some simple calculation, e.g. tight-binding or other, to illustrate the authors’ approach? As is well-known several complications might arise when putting a new method to the test.

We thank the referee for their time and care in reviewing our manuscript. We do agree with the referee that model calculations implementing our formalism would provide some physical insight. However, because ELWFs are central to our formalism, and because such functions are not provided in simple tight-binding models appearing in the literature (except, e.g., approximating them as delta functions), there is no simple calculation we can do at this time. However, we do stress that in our approach, it is only the charge and current densities found from polarization and magnetization fields, and free charge and current densities, that are typically physically accessible. Thus, any such calculation would not provide additional support to our theory. Rather, to support our approach, we apply the general definitions to obtain a number of quantities and compare them to existing expressions in the literature, when possible. In particular, we compare our results with those of Blount and co-workers, and also with the metallic electrical conductivity tensor (for e.g., below Eq. 35). To clarify these issues, we have significantly modified the Abstract, Introduction, and Conclusion sections. To the Introduction we have added the paragraph beginning with “A set {…} satisfying (1) is far from unique…” and modified the two paragraphs that follow. At the end of the Introduction we add the text starting with “Ultimately, when implemented in a metallic crystal, that our general definitions agree …”. In rewriting the Abstract and refining the Conclusion (per the referees later suggestion), we believe our arguments are now more clear.

Please find some further questions and comments below

1) The main results of the paper seem to be the definition of the polarization in Eq. (26) and the definition of the magnetization in Eq. (27). Is this indeed the case?

In our opinion, the main results are the expressions for the unperturbed polarization (26) and magnetization (29), as well as the induced quantities (30), (34), and (35). These expressions are, in fact, obtained from general definitions previously presented in Ref. [13], which we here apply to a metallic crystal initially occupying its zero-temperature ground state. In particular, the induced quantities are understood as a generalization of the previously identified “bound” and “free” contributions to induced currents in metals found by Blount and others.

2) The only thing that can be measured in experiment is the change of polarization which is fully defined in terms of the time integral of the current density (assuming there is no magnetization). This works equally well for insulators and for metals. So, what is the importance of defining an absolute polarization which is a gauge-dependent quantity? (see also point 7 below)

Generically, what can be measured in experiment is induced charge and current densities, which here also involve “free” contributions. This contrasts the insulating case. In the special case that polarization and magnetization happen to be gauge-invariant, they too may also be accessible physically, but this is not generally the case. To clarify this issue we added the paragraph beginning with “A set {…} that satisfies (1) is far from unique…”, add the sentence “Here, polarization and magnetization fields, and free charge and current densities serve as intermediary quantities that aid calculation and provide physical insight, but in general only the appropriate combinations that lead to the charge and current densities have direct physical significance.” and we modified the paragraph beginning with “For an unperturbed trivial insulator occupying its zero-temperature ground state…”.

3) In the abstract the authors write “If instead one takes the view that the electronic polarization is more fundamentally related to the existence of a complete set of exponentially localized Wannier functions, a definition is always admitted.” I do not see what is fundamental about the existence of a complete set of exponentially localized Wannier functions (ELWF). They might be very useful in practice for numerical calculations and/or for interpretation of the results but what is fundamental about them?

We do not mean that ELWFs are inherently fundamental to the electronic dynamics, just as notions of polarization, magnetization, etc., are not. But, our perspective is that, were one to seek a definition of the former, ELWFs may be used as the basis of such definition. We have reworded the abstract to avoid any misleading statements and believe the modified text explicitly referenced in our response to 3) also clarifies this.

4) The authors use site-based quantities such as a site density and site current density but they are never explicitly defined. Could the authors please add these definitions (or point to the appropriate equations in earlier works)? Are the definitions general or are they contingent on the choice of using ELWF?

As mentioned in the text surrounding Eq. 24, 25, these definitions were presented in Ref. [13]. For convenience we add the reference equation numbers to this text. In general, one could use any complete set of sufficiently localized functions to define the polarization, magnetization, etc. However, a set of ELWFs is the most convenient, and indeed it is the case that any set of ELWFs will span the same space as that of those other purported functions (by definition of “complete” here). Then, such basis sets would be related by a gauge transformation and thus any experimentally accessible results would likely be independent of this choice, since they need be gauge invariant. Then, while our definitions of polarization, magnetization, etc., are in general sensitive to the particular choice of ELWFs – they are gauge dependent – any such choice is valid, and physically accessible quantities should be independent of this choice.

5) In the introduction it is written that a lattice sums are performed to give the microscopic polarization and magnetization. Lattice sums are generally conditionally convergent which could lead to convergence problems in practical calculations. Could the authors comment on this?

Since the summand “site” microscopic fields, which involve matrix elements of Wannier functions associated with the reference lattice site and all others, are well-localized about that reference lattice site with which they are associated (see text below Eq. (46,56,66) of Ref. [13]), if one is interested in the behaviour of a given microscopic field near a point in space, in practice, only the site quantities associated with lattice sites “near” that point will contribute. So, in practice, such infinite lattice sums could be truncated when considering a small region of space.

6) Also in the introduction, the authors write “One reason is that usual calculations made in minimal coupling can require the identification of sum rules to show properly behaved results at low frequencies - especially if nonlinear optical response is calculated, which is a future direction for this work - and that is not a difficulty with the calculations presented here, since the response is calculated as due to electric and magnetic fields.” In the linear response the problem of divergencies at low frequencies have been addressed in the literature and several solutions have been proposed (PRB 82 035104, PRB 95 155203). Indeed, I think the use of sum rules is an elegant approach to avoid divergencies.

We tend to agree with this comment. We do note that, when considering more complicated susceptibility tensors, such sum rules are not typically obvious and might take substantial amounts of time to find. This method avoids such practical issues.

7) Just below the authors write “A second reason is that in an insulator there is a clear physical significance to the response of the polarization to applied fields, as has been demonstrated within the “modern theory,"” Could the authors explain what this physical significance is?

The physical significance here is that, in an ordinary insulator, the entire electronic response to an electric field is due to a modification of the electric polarization about each lattice site, and this is gauge invariant.

8) Why do the authors neglect local-field corrections which can be very important? Moreover, no calculations are performed in this work, everything is purely formal, so why neglect them?

We agree that such corrections can be important. In this first work, we neglect such corrections to give room to fully analyze this more simple case, which is indeed not so very simple.

9) The authors write “Consideration of the phenomena related to spatially-varying electromagnetic fields is left for future work.” but one page further they write “The primary difference between our approach and those is that this investigation is a limiting case of a more general framework within which spatial and temporal variation of electric and magnetic fields can be into account; that is not the case in earlier works.” If this is the primary difference it would seem important to not defer spatially-varying electromagnetic fields for future work but to include them here.

We agree with the referee that such an investigation would be very interesting. We do postpone calculations considering spatially-varying fields for future work, however, to avoid further lengthening of this manuscript, and to maintain some focus on the primary takeaways from these initial calculations. In this manuscript, we do consider temporally varying fields, which are already beyond the domain of the “modern theories.”

10) In the conclusions the authors write “A positive feature of the approach implemented here is that it captures the complete finite-frequency response of a metallic crystal; there is no need to implement distinct approaches to calculate the anomalous Hall and Drude contributions to J(1).” Could the authors compare to PRB 86 125139 which also seems to include all contributions. Drude contributions are also included in PRB 74 245117 but not anomalous Hall contributions because time-reversal symmetry is assumed. However, I would expect that also this contribution is present if this assumption is removed.

We do not claim this feature is unique to our method, only that it is a supporting aspect of it. We emphasize that consistency with works such as those referenced by the referee add support to the definitions we employ, as we arrive at the expected result via a new method. We have added text to the Introduction and refined the Conclusion to make this point more clear. For example, at the end of the Introduction we add the text beginning with “Ultimately, when implemented in a metallic crystal, that our general definitions agree with past work…”.

11) In appendix B the authors write “Although most derivations [43] work with Bloch functions from the onset, some issues related to the position operator can be avoided if one works, at least initially, in the Hilbert space containing ELWFs, the space of square-integrable functions, where the usual position operator is well-defined.” The usual position operator is not compatible with periodic boundary conditions (PBC) and, therefore, ill-defined whenever PBC are applied. Therefore, I do not understand why the authors claim that by invoking ELWFs the usual position operator is well-defined. Could the authors explain this point in more detail? I note that recently a position operator has been defined that is compatible with PBC (PRB 99 205144, PRB 105 235201).

In a subspace of the Hilbert space of square-integrable functions, the position operator is well-defined (e.g., the usual position expectation value in a space of atomic wavefunctions is well-defined). A basis of ELWFs spans such a subspace, while the Bloch functions are not even square-integrable. This allows the position matrix elements in the basis of ELWFs to be well-defined, in particular since they have compact support; See, e.g., Ref [43].

Some minor points

12) The authors write “However, often times, and as we will take to be the case in this paper, a number of Hilbert subbundles of the Bloch bundle are trivializable, …”. What is meant by “trivializable” ?

We thank the referee for pointing this out; the sentence should read “a number of Hilbert subbundles of the Bloch bundle are trivial, …” in that there exists a global trivialization of each. We have corrected this.

13) The conclusion section is almost two pages long. The authors might consider to be more concise.

We heed this advice, in the hope that it aids readability and improves clairity.

14) References [25] and [60] are the same.

This has been corrected.

---

## Round 2 · List of Changes

We have significantly revised the Abstract, Introduction, and Conclusion sections of this manuscript. To clarify confusion and elucidate our findings, we have almost entirely re-worked the Abstract and refined the Conclusion. In the Introduction, we have removed some technical remarks to aid readability, and modified various paragraphs. In particular, we have significantly added the paragraph beginning with "A set {...} that satisfies ..." and modified the two that follow, and also modified that beginning with "Our conclusions and perspectives...".

---

## Round 3 · Referee Report · Anonymous (Referee 1) · 2022-12-17

Report

Dear editor,

I thank the authors for their answers to me questions and comments and for the second revision of the manuscript.
I think the manuscript can now be accepted an article in SciPost Physics.

---

## Round 3 · List of Changes

In response to the referees comments we have made the following changes: - We have added the “mixed” form of (24) above it as an unnumbered equation as well as the text beginning “Although the matrix elements appearing in …” just before the start of Sec. III A. - We have also added the footnote beginning with "For example, in the linear response of a “topologically trivial” insulator, ...". - We have added references to the publications PRB 82 035104, PRB 95 155203, and PRB 86 125139 - We have modified the sentence beginning with "To simplify these initial considerations we here neglect local field corrections, ...".

---

## Editorial Decision

published